# Flexibility-conditioned protein structure design with flow matching

Vsevolod Viliuga [* 1 2 3]   Leif Seute [* 1 3 4]   Nicolas Wolf [1 3 4]   Simon Wagner [4]
Arne Elofsson [2]   Jan Stühmer [1 5]   Frauke Gräter [3 1 4]

## Abstract

Recent advances in geometric deep learning and generative modeling have enabled the design of novel proteins with a wide range of desired properties. However, current state-of-the-art approaches are typically restricted to generating proteins with only static target properties, such as motifs and symmetries. In this work, we take a step towards overcoming this limitation by proposing a framework to condition structure generation on flexibility, which is crucial for key functionalities such as catalysis or molecular recognition. We first introduce BackFlip, an equivariant neural network for predicting per-residue flexibility from an input backbone structure. Relying on BackFlip, we propose FliPS, an SE(3)-equivariant conditional flow matching model that solves the inverse problem, that is, generating backbones that display a target flexibility profile. In our experiments, we show that FliPS is able to generate novel and diverse protein backbones with the desired flexibility, verified by Molecular Dynamics (MD) simulations. FliPS and BackFlip are available at https://github.com/graeter-group/flips.

## 1. Introduction

The past few years have arguably marked the golden age of *de novo* protein design. Rapid advancement of deep learning-based approaches and their application to the protein structure prediction problem have enabled an unprecedented level of modeling accuracy (Jumper et al., 2021; Dauparas et al., 2021; Lin et al., 2022). This, in turn, paved

*Equal contribution [1]Heidelberg Institute for Theoretical Studies, Heidelberg, Germany [2]Dept. of Biochemistry and Biophysics at Stockholm University and Science for Life Laboratory, Stockholm, Sweden [3]Max Planck Institute for Polymer Research, Mainz, Germany [4]IWR, Heidelberg University, Heidelberg, Germany [5]IAR, Karlsruhe Institute of Technology, Karlsruhe, Germany.

*Proceedings of the 42$^{nd}$ International Conference on Machine Learning*, Vancouver, Canada. PMLR 267, 2025. Copyright 2025 by the author(s).

the way for more controllable and precise design of protein structures with specific functional properties. For example, proteins with on-demand chemical reactivities or binding propensities are of particular interest for biotechnological applications (Lovelock et al., 2022), therapeutics development (Bennett et al., 2024) and even sustainability problems, such as plastic degradation (Chen et al., 2022).

Only recently, several deep generative models for protein structure design based on diffusion (Song et al., 2021) or flow matching (Lipman et al., 2023; Albergo & Vanden-Eijnden, 2022) have successfully demonstrated high biophysical consistency of generated samples (Watson et al., 2023; Bose et al., 2024; Yim et al., 2023a). The molecular context that such state-of-the-art models can be conditioned on is typically restricted to static properties like the presence of a certain motif in the final structure, adherence to specific symmetry groups or binding to target proteins or ligands (Ingraham et al., 2023; Krishna et al., 2024; Yim et al., 2024; Lin et al., 2024; Huguet et al., 2024). Protein structures generated by deep learning methods tend to be highly thermostable, (Watson et al., 2023; Zambaldi et al., 2024), which suggests high degree of structural rigidity (Scandurra et al., 1998), and incorporating flexibility into specific parts of a backbone remains a great challenge. Proteins, however, must possess dynamic behavior in order to encode function such as chemical reactivities or specific recognition of binding partners. For example, enzymes have complicated free-energy landscapes (Benkovic et al., 2008) and traverse from one conformation to another in their catalytic cycle, which includes loop opening and closure, concurrent domain movements and local folding and unfolding events (Miller & Benkovic, 1998; Liao et al., 2018; Zinovjev et al., 2024). Also molecule-specific protein binders require local structural flexibility in order to engage in a complex with DNA, RNA or ligand molecules (Glasscock et al., 2023; Pacesa et al., 2024; Guo et al., 2024). Since protein structure defines protein function, the limited structural diversity and high rigidity of proteins designed with state-of-the-art generative models impede the exploration of a vast potential functional space.

**Main contributions**   In this work, we propose a framework for generating protein structures conditioned on structural flexibility relying on two key innovations. We first

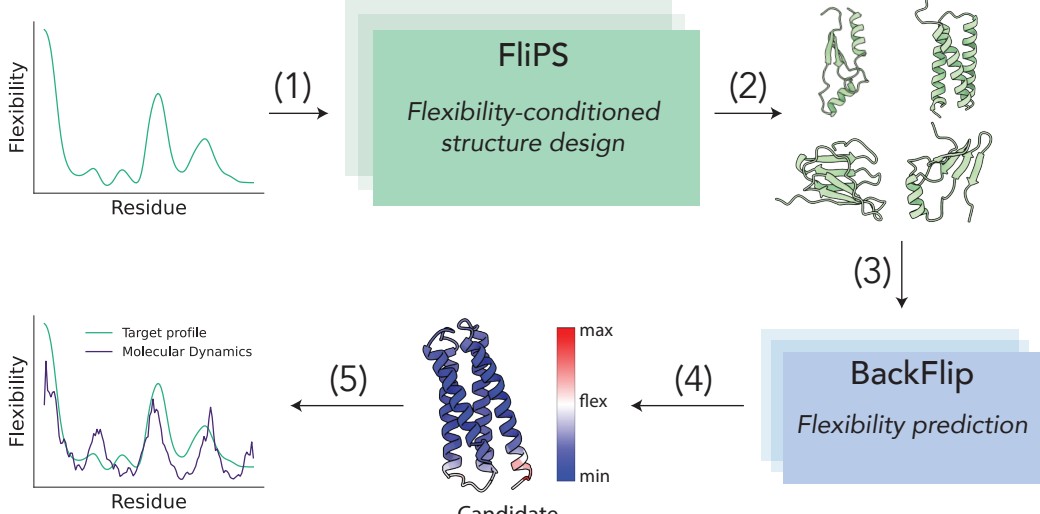

*Figure 1.* Flexibility-conditioned protein backbone design. Given a target flexibility profile (1), the conditional generative model FliPS is used to sample several candidate backbones (2). Flexibility profiles of backbones are predicted with BackFlip (3) and the most promising candidate is selected (4). The flexibility of the protein can then be validated in a Molecular Dynamics simulation (5).

introduce **(1) BackFlip** – an equivariant neural network trained to predict per-residue flexibilities of a given protein backbone entirely independent of sequence information. Utilizing BackFlip for large-scale flexibility annotation and an auxiliary loss, we propose **(2) FliPS** – a generative model for protein structure conditioned on per-residue flexibilities, defined as magnitude of local fluctuations in Molecular Dynamics (MD) simulation. FliPS is an extension of the recent Geometric Algebra Flow Matching (GAFL), an SE(3)-equivariant flow matching model for protein structures with diverse secondary structure composition of the generated backbones (Wagner et al., 2024). We combine the two proposed models and introduce a flexibility conditioning pipeline, in which backbones are generated with the conditional model FliPS and ranked with BackFlip (Fig. 1).

In our experiments we demonstrate that the proposed flexibility-conditioning pipeline generates novel and diverse protein structures that display a given flexibility profile, verified by MD simulations on the timescale of 300 ns.

### 1.1. Related work

Generative models for protein structure have been conditioned on static structural features, such as symmetry groups or motifs (Watson et al., 2023; Yim et al., 2024) and on protein or ligand targets (Watson et al., 2023; Ingraham et al., 2023; Krishna et al., 2024). Deep learning-based tools have been successfully combined with classical modeling tools in the past to design dynamic protein structures, such as allosterically switchable assemblies (Pillai et al., 2024) or pH-responsive filaments (Shen et al., 2024), without explicitly harnessing intrinsic flexibility. Only few works attempted incorporating dynamics into generative models for protein

design. (Komorowska et al., 2024) propose conditioning a pre-trained diffusion model on normal modes that approximate local harmonic movements around an equilibrium state. Instead of learning a conditional model, gradients of an analytical loss on normal modes are used for guidance upon inference. (Kouba et al., 2024) introduce the backbone flexibility prediction model FlexPert-3D that relies on embeddings from a large protein language model (pLM), which we evaluate in Tab. A.1. They propose a framework to condition sequence generation models to better align with a target flexibility, leaving the structure invariant. We discuss relevant works in more details in Appendix A.1.

## 2. Background

### 2.1. Flow Matching for protein structure generation

**Riemannian Flow Matching** The goal of flow matching (Lipman et al., 2023; Tong et al., 2024) is to sample from an unknown distribution $p_1 : \mathcal{M} \to [0, 1]$ on the data domain $\mathcal{M}$ by learning a flow $\phi_t : [0, 1] \times \mathcal{M} \to \mathcal{M}$ that transforms a simple prior distribution $p_0$ to the target $p_1$ via the pushforward $[\phi_1]_* p_0 = p_1$. The flow is parametrized by a learnable, time dependent vector field $v(x, t) : \mathcal{M} \times [0, 1] \to \mathcal{T}_x \mathcal{M}$ on the tangent space $\mathcal{T}_x \mathcal{M}$ according to the flow ODE,

$$\frac{\mathrm{d}}{\mathrm{d}t} \phi_t(x) = v(\phi_t(x), t), \quad \phi_0(x) = x, \qquad (1)$$

which is numerically integrated during inference. Lipman et al. (2023) have shown that the vector field $v$ can be learned by defining trajectories between samples $x_0 \sim p_0$ and $x_1 \sim p_1$ and regressing on the tangent vectors. A common choice for the trajectories are geodesics (Chen & Lipman, 2024). For more details, we refer to Yim et al. (2023a).

**Protein backbone representation** Many state-of-the-art models represent the spatial structure of a protein backbone of $N$ residues as a sequence of $N$ rotations and translations, that is as rigid-body frames $T \in \mathcal{M} \equiv \mathrm{SE}(3)^N$ (Jumper et al., 2021; Yim et al., 2023b). In diffusion and flow matching for protein structure, it is common practice to learn the flow vector field $v$ (Eq. 1) by predicting a de-noised target structure $\hat{T}_1(T_t, t)$ from the intermediate structure $T_t$ with a neural network and then calculating a prediction for $v$ as the tangent vector of the geodesic between $T_t$ and $T_1$. Yim et al. (2023a) have shown that, for the manifold $\mathrm{SE}(3)^N$, the tangent vectors to geodesics $T \equiv (x, r)$ connecting $T_t \equiv (x_t, r_t)$ and $T_1 \equiv (x_1, r_1)$ can be calculated as

$$\frac{\mathrm{d}x}{\mathrm{d}t} = \frac{x_1 - x_t}{1 - t}, \quad \frac{\mathrm{d}r}{\mathrm{d}t} = \frac{\log_{r_t}(r_1)}{1 - t}, \quad (2)$$

where the frames $T \in \mathrm{SE}(3)^N$ are decomposed into translational part $x \in \mathbb{R}^{3N}$ and rotational part $r \in \mathrm{SO}(3)^N$.

**Model architecture** In order to learn the function $T_1(T_t, t)$ as described above, Yim et al. (2023b) introduce a neural network architecture based on the structure block of AlphaFold2 (Jumper et al., 2021). The input features include the noised frames $T_t$, their pairwise spatial distances, positional encodings of absolute and relative sequence positions, and the flow matching time $t$. These features and the frames defining the protein structure are updated consecutively in a series of six blocks that use Invariant Point Attention (IPA), introduced in Jumper et al. (2021), as a central component.

Wagner et al. (2024) propose replacing the point-valued features of IPA with features embedded in the Projective Geometric Algebra (PGA), yielding a geometrically expressive latent representation for protein structure. They integrate this extension, termed Clifford Frame Attention (CFA), into the flow matching framework FrameFlow (Yim et al. (2023a)), and call the resulting model GAFL.

### 2.2. Metrics of protein flexibility

Proteins display dynamic behavior to perform functions, such as catalysis, association, or regulation (Teilum et al., 2009). To quantify protein flexibility, several metrics are being used in the field, based on experiments or simulations.

**B-factor** Typically, structures of proteins resolved in experiments contain data on the deviation of atoms from their mean position, referred to as a B-factors (Sun et al., 2019). Since B-factors strongly depend on the physical conditions under which the experiment was conducted, such as temperature and the radiation source (Eyal et al., 2005), comparing B-factors of two independently determined protein structures is challenging and requires a series of normalizations (Djinovic-Carugo & Carugo, 2015).

**pLDDT** Recently, the confidence measure LDDT predicted by AlphaFold2 (pLDDT) has been suggested as a proxy for the flexibility and disorder in proteins (Ruff & Pappu, 2021; Alderson et al., 2023). An advantage of pLDDT values over B-factors is that they can be obtained *in silico*, without requiring wet-lab experiments.

**RMSF** One of the most established metrics for assessing per-residue flexibility in a protein is the Root Mean Square Fluctuation (RMSF) derived from Molecular Dynamics (MD) simulations or in-solution NMR experiments. RMSF measures positional deviation of each residue after aligning protein states to a reference conformation. However, there is no wide consensus on the choice of the reference conformation. Moreover, RMSF in its conventional definition relies on the global superposition of the entire protein, causing a non-locality that can lead to ambiguities and artifacts, as we illustrate in Section A.11 and Fig. A.3.

**Local RMSF** Tackling the above-mentioned issues of global RMSF, we propose a generalization in which the fluctuations of a residue are measured with respect to its local surrounding instead of the whole protein. We introduce a fluctuation scale $S$ that quantifies the number of neighbors used for alignment of each residue, such that $S = \infty$ corresponds to the conventional global RMSF. We define the local flexibility $\xi$ of residue $i$ in a set of conformations $\mathcal{C}$ and reference conformation $x^{(\mathrm{ref})}$ as

$$\xi_i \equiv \sqrt{\frac{1}{|\mathcal{C}|} \sum_{x \in \mathcal{C}} \left| \left( T_{\mathrm{Align}}^{(S)} \circ x \right)_i - x_i^{(\mathrm{ref})} \right|^2}, \quad (3)$$

where the transformation $T_{\mathrm{Align}}^{(S)}$ *locally aligns* residues in the window $W \equiv \{i - \frac{S}{2}, \ldots, i + \frac{S}{2}\}$,

$$T_{\mathrm{Align}}^{(S)} \equiv \mathrm{argmin}_{T \in \mathrm{SE}(3)} \left\| T \circ x_W - x_W^{(\mathrm{ref})} \right\|_2. \quad (4)$$

In our experiments, we use $S = 12$ neighboring residues in the sequence. In order to remove potential biases induced by the choice of reference conformation, we randomly choose $N_{\mathrm{ref}} = 10$ reference conformations from the MD trajectory. This results in $N_{\mathrm{ref}}$ local RMSF values for each residue, across which we take the median. We visualize the difference to global RMSF in Fig. A.4.

## 3. BackFlip: Backbone flexibility predictor

In this section, we introduce BackFlip (**Back**bone **Fl**ex**i**bility **P**redictor), an SE(3)-equivariant neural network capable of predicting per-residue protein flexibility from the backbone structure alone. Consequently, **BackFlip does not rely on the protein's sequence** and, in particular, **does not use any evolutionary information** and no pre-trained protein language model for its prediction, making it especially useful for *de novo* protein design applications.

## 3.1. Architecture and training

BackFlip's architecture is based on the GAFL architecture, an SE(3)-equivariant transformer with Clifford Frame Attention (CFA) introduced in (Wagner et al., 2024) as extension of AlphaFold's invariant point attention (IPA) (Jumper et al., 2021). We represent a protein backbone as an element of $SE(3)^N$, that is as set of Euclidean frames, as in (Jumper et al., 2021). Node features are expressed in the local coordinate systems given by the frame of the respective residue and transformed into the target frame during message passing. This not only ensures SE(3)-equivariance but also encodes geometric information about relative displacement and orientation of the residues. While in GAFL and AlphaFold2 the backbone structure is updated after each message passing block, the regression task performed by BackFlip does not require structural updates. Instead, we extract intermediate node features after $n_{\mathrm{cfa}}$ blocks of CFA, feed them through a node-wise Multilayer Perceptron (MLP) and map the output to the range $[0, \xi_{\max}]$ using a scaled sigmoid,

$$\xi_i = \xi_{\max} \sigma \left( \mathrm{MLP} \left( h_i \right) \right) , \tag{5}$$

where the scalar value $\xi_i$ represents the flexibility of residue $i$. For our experiments, we choose $n_{\mathrm{cfa}} = 4$, a node and edge embedding size of 64 and 32, respectively, and a maximum flexibility of $\xi_{\max} = 5\,\text{Å}$. With these hyperparameters, BackFlip has 0.68 M parameters, compared to 16.7 M parameters of GAFL, which has six blocks of CFA and around four times higher embedding dimensions. A schematic representation of the architecture is depicted in Fig. A.1.

We train BackFlip with a mean squared error loss on per-residue flexibilities of backbones from the ATLAS dataset (Vander Meersche et al., 2024). ATLAS is comprised of MD simulations of 1390 structurally non-redundant proteins conducted as three replicas for a total length of 300 nanoseconds (ns), from which we calculate ground truth per-residue flexibilities as local RMSF (see Sections 2.2 and A.11). We filter ATLAS for proteins up to the length of 512 residues, resulting in a total of 1294 proteins, which we split in 1035 backbones for training, 130 for validation and 129 for testing (further details in Appendix A.2).

### 3.2. Performance on unseen proteins

In order to test BackFlip's performance on unseen proteins, we evaluate it on held-out test proteins of the ATLAS dataset. For that, we pass equilibrium experimental structures as input to BackFlip and predict their local RMSF profiles. Since AlphaFold2's confidence measure pLDDT (Jumper et al., 2021) has been suggested as measure for disorder and flexibility before (Ruff & Pappu, 2021; Akdel et al., 2022; Alderson et al., 2023), we compare BackFlip predictions with pLDDT. We also investigate how well experimental B-factors reflect local RMSF.

In Tab. 1 we report the correlation and mean absolute error (MAE) of BackFlip's predicted local RMSF compared with the value obtained from two of the three MD trajectories per protein in ATLAS. In order to quantify the noise present in the ground truth, which is due to the trajectories only covering finite time and thus not sampling the free energy landscape extensively, we hold out the third MD trajectory and report its correlation and MAE to the two other trajectories. We find that BackFlip outperforms any baseline considered and, indeed, reflects the shape of the local RMSF profiles with a Pearson correlation coefficient of 0.8 on the 129 unseen proteins from ATLAS. Not only the shapes but also the amplitudes of the profiles are matched, indicated by an MAE of 0.17 Å. BackFlip's remaining error can be in part attributed to the noise present in the ground truth since MD only performs slightly better, with a correlation of 0.84 and an MAE of 0.14 Å. We find that pLDDT only moderately displays trends of local RMSF and experimental B-factors correlate poorly with MD-derived local flexibility. For both pLDDT and B-factors, a direct conversion to local or global RMSF is not available, thus MAEs are not reported.

In addition to the ATLAS test set proteins, we sample protein backbones that match the length distribution of the ATLAS dataset with FrameFlow (Yim et al., 2024) and RF-diffusion (Watson et al., 2023), respectively, and investigate BackFlip's performance on a total of 100 designable *de novo* proteins (Appendix A.6). We conduct three 100 ns long MD simulations per protein (Appendix A.8) and compute ground truth local RMSF profiles as for the ATLAS dataset. Also in this setting, BackFlip demonstrates strong performance in terms of correlation and MAE, indicating generalization beyond naturally occurring proteins (Tab. 1).

We also compare BackFlip's performance with FlexPert-3D (Kouba et al., 2024), a recent backbone flexibility prediction model trained on the ATLAS dataset, and find that Back-Flip is either on-par with or outperforms FlexPert, which relies on a large pre-trained protein language model, on both natural and *de novo* proteins (Appendix A.5).

In Fig. A.2, we show that BackFlip's predictions are consistent for proteins of different sizes and visualize three selected backbones with color-coded flexibility profiles (Fig. 2A, B). We find that loops and turns are identified as the most flexible regions (Fig. 2B), which is expected since these structural elements typically have less stabilizing non-covalent interactions and thus can exhibit higher conformational heterogeneity. The model also identifies rigid regions, such as $\alpha$-helices and $\beta$-sheets residing in the tightly packed core, as the least flexible regions. Notably, the model also reliably captures less obvious alterations in local flexibility, such as shorter secondary structure elements with intermediate mobility (e.g. residues 80-100 in 4zi3D, right panel).

*Table 1.* Performance of BackFlip for the local RMSF derived from two 100 ns long MD simulations on unseen proteins from ATLAS and 100 *de novo* proteins. We also evaluate the flexibility proxies B-factor and negative pLDDT. We report the Pearson correlation coefficient across all residues (Global $r$), the mean absolute error (MAE) and inference time for the 300-residue protein 7c45A on a single NVIDIA A100 without batching. Errors are computed by bootstrapping the $2N_{\text{ref}} = 20$ ground truth RMSF profiles 100 times (see Section A.11).

| | ATLAS test set | | De novo proteins | | |
|---|---|---|---|---|---|
| | Global $r$ ($\uparrow$) | MAE [Å] ($\downarrow$) | Global $r$ ($\uparrow$) | MAE [Å] ($\downarrow$) | Time [s] ($\downarrow$) |
| MD (Ground Truth) | 0.84 (0.00) | 0.14 (0.00) | 0.80 (0.01) | 0.10 (0.00) | $\mathcal{O}(10{,}000)$ |
| B-factor* | 0.16 (0.01) | - | - | - | - |
| Negative pLDDT | 0.54 (0.00) | - | 0.48 (0.00) | - | 118 |
| **BackFlip** | 0.80 (0.00) | 0.17 (0.00) | 0.73 (0.00) | 0.11 (0.01) | 0.6 |

*Computed for proteins where available.

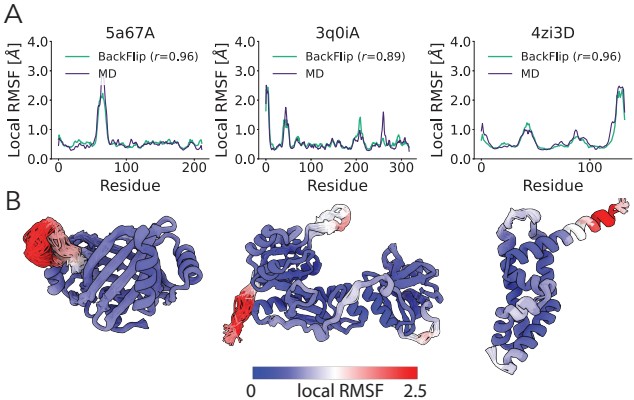

*Figure 2.* BackFlip accurately predicts protein backbone flexibility. (A) Flexibility profiles of three selected proteins from the test set, derived from Molecular Dynamics (ground truth) and predicted by BackFlip along with their Pearson correlation. (B) Visualization of the corresponding proteins and their locally aligned MD-ensembles, colored by the BackFlip-predicted local RMSF.

## 4. FliPS: Flexibility-conditioned protein structure generation

In this section we introduce a flow-based generative model FliPS (Flexibility-conditioned Protein Structure generation,) capable of generating protein structures that display a given target flexibility profile. The model is based on Geometric Algebra Flow Matching (GAFL) (Wagner et al., 2024) – an unconditional generative model for protein backbone design with strong performance in designability and secondary structure diversity of sampled backbones. In order to condition on structural flexibility, we propose a modification of GAFL by introducing a flexibility embedding, a flexibility auxiliary loss, and a flexibility-masking procedure during training. We term the resulting conditional model FliPS and demonstrate that conditional sampling yields protein backbones that display flexibility profiles closely resembling the target profile while being structurally diverse. As alter-

native to training a conditional model, we also propose a training-free BackFlip-guidance (BG) strategy for flexibility conditioning that can be integrated into existing frame-based unconditional models like RFdiffusion (Watson et al., 2023). We also introduce a screening procedure based on BackFlip to find naturally occurring or generated protein backbones that display the desired flexibility profile.

### 4.1. Method

In order to sample from the conditional probability distribution $p(T|\xi)$ of protein structures $T \in \text{SE}(3)^N$ with given per-residue flexibility profile $\xi \in \mathbb{R}_+^N$, we train a conditional flow matching model by approximating the conditional flow vector field $v(x, t, \xi)$ defined analogously to Eq. 1,

$$\frac{\mathrm{d}}{\mathrm{d}t}\phi_t(T|\xi) = v(\phi_t, t, \xi), \quad \phi_0(T|\xi) = T. \quad (6)$$

**Training objective** During training, we sample points in time $t \sim \mathcal{U}([0, 1])$, prior structures $T_0 \sim p_0$ and target structures with flexibility profiles $(T_1, \xi_1) \sim p_1$ from the training set. We randomly mask parts of $\xi_1$, or the whole flexibility profile, in order to prevent memorization and retain the capability of unconditional structure generation (Appendix A.9). We calculate the intermediate structure $T_t$ along the geodesic connecting $T_0$ and $T_1$ and predict the vector field $\hat{v}(T_t, t, \xi_1)$ with the model, using the parametrization in Eq. 2. We regress on the ground truth vector field $v$,

$$\mathcal{L} = \mathbb{E}_{t, T_0 \sim p_0, (\xi_1, T_1) \sim p_1} \left[ \left\| v - \hat{v}(T_t, t, \xi_1) \right\|^2 + l_{\text{aux}} \right] \quad (7)$$

using the norm and heuristic auxiliary loss function $l_{\text{aux}}$ proposed in (Yim et al., 2023b). We extend $l_{\text{aux}}$ by a novel flexibility loss as described below. The key difference to unconditional training is that we pass the flexibility profile $\xi_1$ of the target structure to the model $\hat{v}$, implicitly learning the relationship between flexibility and structure.

**Flexibility embedding** In order to pass the flexibility profile $\xi$ to the model, we embed the per-residue flexibility

as additional node input feature. For this, we divide the flexibility into 8 bins with maximum value of $\xi_{max} = 3\,\text{Å}$.

**Flexibility auxiliary loss**   We propose an additional auxiliary loss term that explicitly penalizes the generation of structures with deviating flexibility profile during training. To this end, we use the predicted target structure $\hat{T}_1(T_t, t, \xi_1)$, obtained via Eq. 2, to retrieve predicted flexibilities for all residues $\xi_{BF}(\hat{T}_1)$ by applying BackFlip to the predicted target structure. Importantly, as the predictor $\xi_{BF}$ of BackFlip is differentiable, we can construct a differentiable auxiliary loss term that penalizes a deviation of predicted and target flexibility profile via the mean square error loss and add this term to the auxiliary loss in Eq. 7,

$$l_{aux} = l_{aux,\,FD} + \frac{\lambda_{flex}}{N}\left\| \xi_1 - \xi_{BF}\left(\hat{T}_1\right)\right\|^2, \qquad (8)$$

where $l_{aux,\,FD}$ is the original auxiliary loss from FrameDiff (Yim et al., 2023b). Notably, this is only possible because BackFlip is differentiable with respect to the backbone structure and does not rely on the protein sequence as input.

**BackFlip guidance (BG)**   As an alternative to training a model to approximate the flexibility-conditioned flow vector field $v(T_t, t, \xi)$ directly, as described above, we also propose BackFlip guidance (BG) – a training-free approach based on BackFlip for guiding an unconditional model upon inference. Inspired by classifier-guidance (Dhariwal & Nichol, 2021), we add the gradient of the deviation between BackFlip-predicted flexibility $\xi_{BF}$ and target flexibility $\xi$ of the intermediate structure $T_t$ at flow matching time $t$,

$$\hat{v}_{cond}(T_t, t, \xi) = \underbrace{\hat{v}(T_t, t)}_{\text{unconditional}} - \underbrace{\eta\nabla_{T_t}\left\| \xi - \xi_{BF}(T_t)\right\|^2}_{\text{cond. guidance term}}, \quad (9)$$

where the BG scale $\eta$ is a hyperparameter. In an ablation study (Tab. A.4), we find that BackFlip guidance with FrameFlow as unconditional model underperforms the conditional model FliPS, which we describe in more details in Appendix A.13.

**BackFlip screening (BFS)**   After (flexibility-conditioned) backbone generation, we identify protein backbones that best display the target flexibility profile $\xi_{ref}$ using a flexibility screening strategy relying on BackFlip. We note that flexibility-screening of *de novo* backbones benefits from an efficient model for flexibility prediction that is independent of the sequence in order to apply the screening before the expensive refolding pipeline (see e.g. (Yim et al., 2023b)).

For a set of backbones, we predict the flexibility profiles with BackFlip and compute (i) the Pearson correlation $r$ and (ii) the mean absolute error (MAE) between the predicted profile $\xi$ and the profile of interest $\xi_{ref}$. We assign a profile

similarity score $s$ by combining the two metrics in order to evaluate both the shape and the amplitude of flexibility,

$$s(\xi, \xi_{ref}) = w_{corr}\, r(\xi, \xi_{ref}) - w_{mae}\, \text{MAE}(\xi, \xi_{ref}), \quad (10)$$

where we choose the weights $w_{corr} \equiv 1$ and $w_{mae} \equiv 2$.

## 4.2. Experiments

**Training**   We train FliPS on the PDB dataset (Berman et al., 2000) introduced in FrameDiff (Yim et al., 2023b) annotated with BackFlip-predicted residue-level flexibilities[1]. We filter the dataset for lengths between 60 and 512 residues and exclude all structures containing breaks, which results in a set of 22977 proteins. The hyperparameters are chosen as in GAFL (Wagner et al., 2024) and $\lambda_{flex}$ is set to 100. We train the model for a total of 21 GPU days on eight NVIDIA A100 GPUs (details in Appendix A.9.)

**Evaluation setup**   To illustrate that FliPS generates plausible protein structures for a range of different target flexibility profiles, we extensively evaluate it on a set of 10 hand-drawn profiles and on 10 profiles of randomly chosen natural proteins. For each profile, we generate 100 samples per backbone length $N \in \{60, 70, \ldots, 120\}$. We further extend the evaluation to bigger proteins with length $N \in \{200, 215, \ldots, 300\}$, for which we pass another 4 hand-drawn flexibility profiles as a condition. We apply BackFlip screening (Section 4.1) to choose the highest-ranked sample that best reflects the target profile and run three 100 ns long MD simulations to assess how well the target profile is recapitulated in practice.

**Baselines**   Since, to the best of our knowledge, FliPS is the first model for the generation of flexibility-conditioned protein structures, we can not directly compare it to existing approaches. However, we apply BackFlip screening to find backbones with the desired flexibility properties sampled with the established unconditional models FoldFlow2 (Huguet et al., 2024) and RFdiffusion (Watson et al., 2023). For that, we calculate a flexibility similarity score for all generated backbones and pick the highest-scored structures, as described in Section 4.1. We also apply the same screening procedure to 3673 natural proteins from the SCOPe dataset (Fox et al., 2014; Chandonia et al., 2022) comprised of protein lengths between 60 and 128 residues (see Appendix A.9). Additionally, we perform an extensive ablation of BackFlip guidance (Section 4.1) and find that its performance is inferior to the conditional FliPS sampling. Results are reported in Appendix A.13 and Tab. A.4.

**Metrics**   We evaluate the similarity of flexibility profiles by calculating the Pearson correlation coefficient $r$ and the

---

[1]Source code, model weights and the flexibility-annotated dataset are available at https://github.com/graeter-group/flips

mean absolute error (MAE) for each profile, measuring how well shape and amplitudes of the profile are recapitulated. We also report established metrics for protein structure generation that measure structural similarity across samples (diversity), similarity to the naturally existing structures (novelty) and consistency with folding and inverse folding models (scRMSD) (Yim et al., 2023b; Bose et al., 2024). For diversity, novelty and scRMSD, we use the same definition as GAFL (Wagner et al., 2024) (see Appendix A.7).

Notably, since scRMSD depends on the superposition of a generated backbone with the refolded structure, and flexibility induces a fundamental uncertainty in the structure of a protein (Halle, 2002), it can be expected that more flexible proteins have worse scRMSD scores. Indeed, we establish a relation between scRMSD and flexibility on natural proteins from the SCOPe dataset (Fig. 3), which are *designable* by definition, questioning the role of scRMSD if generating flexible backbones is intended. We observe a similar relation between scRMSD and novelty of FrameFlow and RFdiffusion samples (Fig. 3).

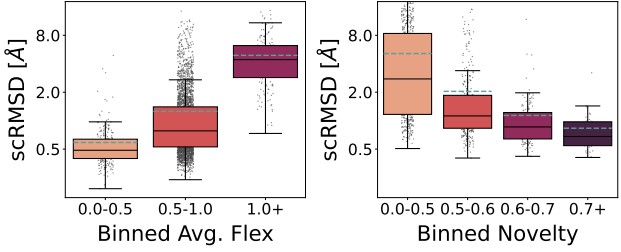

*Figure 3.* Relationship between scRMSD and average local RMSF predicted by BackFlip for proteins from the SCOPe dataset (left) and between scRMSD and novelty for the *de novo* proteins (right) sampled with RFdiffusion and FrameFlow as described in Section 3.2. Vertical axis is log-scaled. Boxes represent 0.25 and 0.75 quantiles. Means are shown as grey dashed lines.

### 4.3. Results

Tab. 2 summarizes the results of the experiment described above on domain-sized proteins and Tab. A.3 on bigger proteins. For each profile, we obtain a total of 700 samples and a single top-ranked sample selected by BackFlip screening. We conduct all-atom MD simulations (see Appendix A.8) of the top-ranked, refolded samples and report novelty of the structure and similarity of the MD-derived flexibility profile to the target (**MD of top samples**).

**FliPS generates structures that recapitulate target flexibility profiles in MD**   We find that FliPS generates backbones whose MD-derived flexibility, indeed, reflects the target profile (Tab. 2 and Fig. 4). For the target profiles considered, the average Pearson correlations $r$ of 0.70, 0.78 for smaller and 0.79 for bigger proteins approach the up-

per bound of around $r_{\mathrm{MD}} \approx 0.8$, set by the fundamental uncertainty between independent (unconverged) MD runs as observed in Tab. 1. Flexibility-conditioning substantially outperforms the flexibility-screening of unconditionally generated *de novo* and natural proteins in similarity to the target profile. We also find that, in terms of novelty, backbones generated with FliPS are on par with backbones sampled by FoldFlow2 and RFdiffusion, which are trained on larger datasets. Remarkably, with only 700 generated samples, FliPS outperforms screening of over 3,500 natural proteins from the SCOPe dataset in terms of capturing flexibility profiles, showcasing the potential of the proposed flexibility conditioning method (Tab. 2). In Fig. 4 we visualize proteins generated with FliPS and their MD-derived flexibility profiles. The samples are composed of both $\alpha$-helices and $\beta$-strands and display local RMSF profiles that mainly follow the shape and magnitudes of the target profiles. Especially for bigger proteins, peak positions and magnitudes are recapitulated by the designs. All other target profiles can be found in Fig. A.5.

**Conditioning increases similarity to the target flexibility** To separate the effect of BackFlip screening from the conditioning in FliPS, we also report the same metrics for all 700 generated samples per profile, that is, without screening for top samples. Since evaluating the flexibility of all structures in MD simulations is prohibitively expensive, we estimate the flexibility with BackFlip and report average values over all profiles (Tab. 2, A.3; **BackFlip on all samples**). We find that for both existing and hand-drawn profiles, protein backbones sampled conditionally with FliPS, on average, reflect the target flexibility profiles substantially better. The strongly improved performance of FliPS over unconditional sampling can be attributed to flexibility conditioning, which is especially prominent for bigger proteins where unconditional samples, on average, do not resemble the flexibility profile at all (Tab. A.3).

**FliPS generates diverse structures with various secondary structure compositions**   Additionally, we investigate how structurally diverse the generated or natural protein backbones are on a set of small proteins from Tab. 2. For that, we pick the 10 top-ranked backbones for each target profile identified with BackFlip screening and report diversity and novelty along with the flexibility correlation in Tab. 3. Note that this is in contrast to Tab. 2, where we apply expensive MD simulation only for one sample per profile. FliPS achieves a maximum pairwise TM score smaller than 0.5 and thus generates structurally distinct backbones, being better or on par with the unconditional baselines. SCOPe proteins display the highest diversity, which can be expected, as this dataset was filtered to contain structurally non-redundant proteins. The experiment also shows that not only one but several backbones with high

*Table 2.* Flexibility conditioned backbone generation for 10 existing and 10 hand-drawn target profiles. For each of the profiles, we generate backbones spanning lengths $N \in \{60, 70, \dots, 120\}$ using FliPS by conditioning on the flexibility profiles. We apply BackFlip screening (BFS) to obtain a best-ranked candidate sample for each profile. For each of the baselines, we sample the same number of structures unconditionally and also apply BFS – or screen the entire SCOPe dataset. For quantifying the effect of conditioning, we also report average values across all sampled backbones without ranking (**BackFlip on all samples**). We report the per-target Pearson correlation ($r$) and the mean absolute error (MAE) to the target profiles, and TM-similarity to the PDB (Novelty). For the best-ranked candidates for each of the profile (**MD of top samples**), we calculate the profile of the generated protein with MD. We report standard deviations obtained by bootstrapping MD profiles as in Tab. 1 and by bootstrapping the set of all generated backbones, respectively.

| | MD of top samples | | | BackFlip on all samples | | |
|---|---|---|---|---|---|---|
| | $r$ ($\uparrow$) | MAE [Å] ($\downarrow$) | Novelty ($\downarrow$) | $r$ ($\uparrow$) | MAE [Å] ($\downarrow$) | Novelty ($\downarrow$) |
| **10 existing profiles** | | | | | | |
| FliPS | **0.70** (0.02) | **0.16** (0.00) | 0.62 (-) | **0.52** (0.00) | **0.15** (0.00) | 0.63 (0.01) |
| RFdiffusion-BFS | 0.58 (0.03) | 0.16 (0.00) | 0.60 (-) | 0.28 (0.00) | 0.19 (0.00) | 0.62 (0.02) |
| FoldFlow2-BFS | 0.45 (0.04) | 0.16 (0.00) | **0.57** (-) | 0.27 (0.00) | 0.18 (0.00) | **0.57** (0.01) |
| SCOPe-BFS | 0.52 (0.02) | 0.19 (0.00) | 1.0 (-) | 0.19 (0.00) | 0.25 (0.00) | 1.0 (-) |
| **10 hand-drawn profiles** | | | | | | |
| FliPS | **0.78** (0.01) | **0.31** (0.01) | **0.55** (-) | **0.56** (0.00) | **0.43** (0.00) | 0.64 (0.03) |
| RFdiffusion-BFS | 0.56 (0.02) | 0.44 (0.00) | 0.60 (-) | 0.13 (0.00) | 0.49 (0.00) | 0.62 (0.02) |
| FoldFlow2-BFS | 0.50 (0.03) | 0.45 (0.01) | 0.57 (-) | 0.08 (0.00) | 0.50 (0.01) | **0.57** (0.01) |
| SCOPe-BFS | 0.61 (0.02) | 0.35 (0.01) | 1.0 (-) | 0.09 (0.01) | 0.48 (0.00) | 1.0 (-) |

*Figure 4.* FliPS samples for two selected existing and hand-drawn target profiles from Tab. 2 (top panel) and for all hand-drawn profiles from Tab. A.3. The generated protein's flexibility is computed from MD simulation. Corresponding protein structure is depicted with sequential coloring (N-terminus blue, C-terminus red).

*Table 3.* Diversity and novelty of flexibility-conditioned backbones from Tab. 2. Metrics are reported as averages over the 10 top-ranked samples for all profiles, selected with BackFlip screening (BFS). We report diversity and novelty based on pairwise TM similarity, scRMSD is computed by aligning the refolded with the generated backbone (see Appendix A.7). We report Pearson correlation $r$ and mean average error (MAE) between the BackFlip predicted and target flexibility profiles. Standard deviations are obtained by bootstrapping all generated samples 10 times.

| | scRMSD | Diversity ($\downarrow$) | Novelty ($\downarrow$) | $r$ ($\uparrow$) | MAE [Å] ($\downarrow$) | Helix pct. | Strand pct. |
|---|---|---|---|---|---|---|---|
| **10 existing profiles** | | | | | | | |
| FliPS | 1.52 (0.01) | 0.36 (0.02) | **0.57** (0.01) | **0.86** (0.01) | **0.09** (0.00) | 0.53 (0.01) | 0.19 (0.01) |
| RFdiffusion-BFS | 0.72 (0.00) | 0.40 (0.02) | 0.61 (0.01) | 0.68 (0.01) | 0.15 (0.02) | 0.78 (0.01) | 0.09 (0.01) |
| FoldfFlow-BFS | 0.72 (0.01) | 0.44 (0.02) | **0.57** (0.01) | 0.67 (0.02) | 0.16 (0.01) | 0.83 (0.01) | 0.02 (0.00) |
| SCOPe-BFS | 0.79 (0.01) | **0.33** (0.02) | 1.0 (-) | 0.72 (0.02) | 0.12 (0.00) | 0.29 (0.01) | 0.33 (0.01) |
| **10 hand-drawn profiles** | | | | | | | |
| FliPS | 2.01 (0.03) | 0.47 (0.02) | 0.60 (0.01) | **0.86** (0.01) | **0.28** (0.00) | 0.60 (0.01) | 0.15 (0.01) |
| RFdiffusion-BFS | 0.74 (0.01) | 0.44 (0.02) | **0.59** (0.01) | 0.66 (0.01) | 0.39 (0.01) | 0.74 (0.01) | 0.11 (0.01) |
| FoldfFlow-BFS | 0.72 (0.01) | 0.45 (0.02) | **0.59** (0.01) | 0.58 (0.02) | 0.43 (0.02) | 0.84 (0.01) | 0.02 (0.00) |
| SCOPe-BFS | 1.96 (0.02) | **0.32** (0.02) | 1.0 (-) | 0.76 (0.02) | **0.28** (0.02) | 0.31 (0.01) | 0.24 (0.01) |

BackFlip-predicted flexibility-similarity can be found, as indicated by the Pearson correlations of over 0.8 between BackFlip-predicted and target profile. Remarkably, while better reflecting flexibility profiles of interest, FliPS backbones also contain higher amount of $\beta$-strands than both RFdiffusion and FoldFlow2, more closely resembling the secondary structure composition of SCOPe proteins.

**Biophysical consistency of protein structures depends on the target profile** To assess consistency of the generated backbones, we calculate scRMSD for the top 10 samples for each profile for the same set of small-sized proteins as described in the previous subsection (Tab 3). We find that, for most profiles, FliPS generates designable backbones with scRMSD below 2 Å. However, backbones generated with FliPS have higher scRMSD values than unconditionally generated samples. We observe that scRMSD strongly depends on the target profile (Fig. A.6), which can be expected since scRMSD is correlated with flexibility and novelty (Fig. 3), both of which can be influenced by the target profile. In this context, we find that even natural proteins from SCOPe display elevated scRMSD values for samples screened for agreement with the hand-drawn profiles.

### 4.4. Discussion

In our experiments we demonstrate that the proposed flexibility-conditioning framework is capable of delivering protein backbones of varying small and bigger lengths that display a range of different realistic and custom hand-drawn flexibility profiles. We note that there is a fundamental uncertainty in MD-derived flexibility due to deviations between simulation runs, as observed in Tab. 1, making the generation of backbones that recapitulate the MD-derived flexibility especially challenging. Importantly, compared to unconditional models augmented by flexibility screening or

guidance, proteins sampled conditionally with FliPS best reflect the desired flexibility, while being structurally non-redundant, novel, and contain both $\alpha$-helices and $\beta$-strands. This finding suggests that FliPS learns a relation between structure and flexibility, exploring a structural space beyond natural proteins in order to find distinct backbones that display the desired flexibility profile.

## 5. Conclusion

We introduce a generative model for protein structure design conditioned on per-residue flexibilities. The proposed flexibility-conditioning framework relies on the structure-based flexibility prediction model BackFlip, enabling large scale flexibility annotation of proteins, a flexibility auxiliary loss and a flexibility screening procedure to find protein backbones that best display a flexibility profile of interest. Our experiments demonstrate that flexibility-conditioning leads to the generation of diverse and novel backbones that indeed display the respective target flexibility profile in Molecular Dynamics simulations. Thus, the proposed framework enables to overcome the current restriction of deep learning-based protein design to static protein structures. While functional protein motions often involve timescales beyond the 300 ns of MD used as flexibility definition in this work, this limitation might be overcome as soon as better ground truth data for flexibility on longer timescales becomes available. Flexibility conditioning can be straightforwardly combined with motif or symmetry conditioning as used to design binding sites or protein assemblies, for which flexibility is required. Since flexibility is crucial for a wide range of protein functionalities, we believe that the proposed method for controlling the flexibility of designed proteins on a per-residue level paves the way for exploring a yet uncharted but highly relevant space of protein structures.

## Acknowledgments

This study received funding from the Klaus Tschira Stiftung gGmbH (HITS Lab). We acknowledge the National Academic Infrastructure for Supercomputing in Sweden (NAISS), partially funded by the Swedish Research Council through grant agreement no. 2021-29 for awarding this project access to the Berzelius resource provided by the Knut and Alice Wallenberg Foundation at the National Supercomputer Centre. The authors acknowledge support by the state of Baden-Württemberg through bwHPC and the German Research Foundation (DFG) through grant INST 35/1597-1 FUGG.

## Impact statement

The societal impact of this work is considered mostly positive, as *de novo* protein design holds the promise to develop drugs against diseases and personalized therapies, which out-weights potential risks (Baker & Church, 2024). While of course security concerns remain, the presented method does not directly enable the design of proteins with a certain biological function, so risk of misuse is low.

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

# A. Appendix

## A.1. Previous work on flexibility flexibility-conditioned protein design

In the past, it was possible to introduce flexibility into the desired protein region with classical modeling tools, such as Rosetta MotifGraft (Alford et al., 2017) along with Rosetta BackRub (Lauck et al., 2010) sampling, Modeller (Šali & Blundell, 1993) or LoopGrafter (Planas-Iglesias et al., 2022). However, these tools require a structure as input to specify the context into which a loop or any given motif should be incorporated. This limits the applicability of these tools to designing scaffolds, but not the novel protein structures, where the starting structure best suited to encode the desired dynamical properties is not known. Another common downside of the classical tools is their relatively high computational cost due to an extensive iterative sampling relying on empirically-parameterized energy functions. Nonetheless, by combining classical tools and deep learning-based approaches, allosterically switchable proteins (Pillai et al., 2024), pH-dependent filamentous self-assembling protein complexes (Shen et al., 2024) and fold-switching proteins (Guo et al., 2024) have been successfully designed without explicitly conditioning on flexibility. In most above-mentioned cases, deep learning, however, is only used for protein structure prediction of designed sequences, not for generating target structures. We see potential for improvement by using generative deep learning with the flexibility conditioning framework proposed in the main text of the paper in design studies described above.

In their work (Komorowska et al., 2024) condition a pre-trained diffusion model on the lowest non-trivial normal modes computed with the Normal Mode Analysis (NMA). Lowest normal modes are obtained from the eigenvectors of the Hessians of the potential energy relying on a force field that assumes the harmonicity of the system, which inherently limits sampling protein structure states to the movements around an equilibrium state. Conditioning is achieved by the gradient guidance whilst gradients are computed from an analytical normal mode loss. Different to their work, we propose training a conditional model akin to classifier-free guidance, whereby the flexibility is derived directly from Molecular Dynamics (MD) simulations. Thus, our approach is not limited to analytical conditioning but handles non-analytical conditions. Since we train our model to approximate a conditional flow, we avoid potential conflicts between unconditional and conditional gradient terms, which might arise in the gradient guidance relying on the analytically computed gradients. We implemented a classifier-guidance strategy, which we term BackFlip-guidance (BG), as an alternative to the conditional FliPS model. However, we found that BG performed worse than conditional sampling and often violated the physicality of the generated backbones, as discussed in A.13.

(Kouba et al., 2024) propose a pipeline to condition protein sequence design on MD-derived flexibility. They introduce flexibility prediction model termed FlexPert-3D, which they use as a scoring model to fine-tune the sequence design model ProteinMPNN (Dauparas et al., 2022) on generation of sequences that better capture the target flexibility observed in MD simulations. FlexPert-3D is a regression model that relies on the pre-trained weights from a protein Language Model and has a trainable CNN head to combine pLM embeddings with Hessian matrices obtained from the Anisotropic Network Model (ANM) computations The approach operates exclusively in sequence space and requires both an input structure and evolutionary information encoded by the pLM embeddings, rendering it fundamentally different from our work. In our experiments we show that BackFlip outperforms FlexPert on both natural proteins from ATLAS and *de novo* proteins in A.5.

## A.2. Training details of BackFlip

In order to train the BackFlip specifically on local flexibility, for each protein in the ATLAS dataset, we first compute local RMSF values as described in Sections 2.2 and A.11. We split the dataset into train, valid, and test set according to the splitting scheme 80:10:10 and limit protein size to 512 residues. An epoch is defined as a single pass through the whole dataset, whereby one of the $N_{ref}$ conformations is randomly selected for each protein. We batch proteins together during training, where each batch contains at most 32 proteins. We train the model on a single NVIDIA A100 GPU and use the value of RMSE between the ground truth and predicted local flexibility scores on the validation set as the early stopping criterion. We then select the best checkpoint based on the RMSE on the validation set and evaluate the model on the test set.

## A.3. BackFlip's architecture

We illustrate a schematic, high-level overview architecture of BackFlip in Fig. A.1.

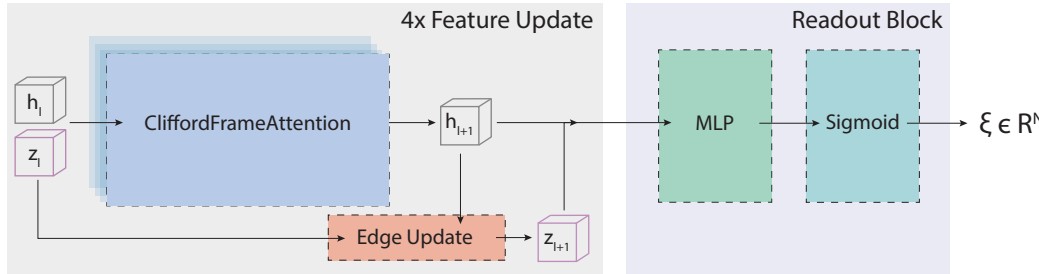

*Figure A.1.* High-level overview of the architecture of BackFlip. Current node $h_l$ and edge $z_l$ embeddings at layer $l$ are updated by the Clifford Frame Attention (CFA) block 4 times. Updated features are input to the Readout Block and processed. The output is a tensor $\xi \in \mathbb{R}^N$, where $N$ denotes the length of a protein. Blocks are indicated with dashed lines.

### A.4. Additional results on BackFlip performance on held-out test set proteins of ATLAS

We find that BackFlip consistently predicts local RMSF profiles for proteins from the ATLAS test set of varying lengths (Fig. A.2A). Also we visualize the distribution of the predicted and ground truth per-residue local RMSF values for the same set of proteins in Fig. A.2B).

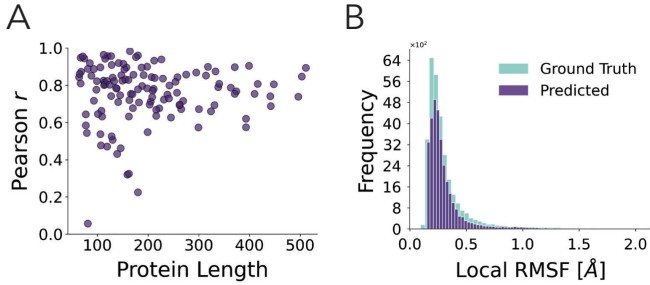

*Figure A.2.* (A) Pearson correlation coefficients of BackFlip predictions on the held-out test set proteins from the ATLAS dataset to the ground truth local RMSF computed from MD simulations as a function of protein length. (B) Distribution of the BackFlip-predicted local RMSF values compared with the distribution of the ground-truth local RSMF for the same set of proteins as in (A).

### A.5. Comparison of BackFlip with FlexPert on the ATLAS and de novo datasets

We compared the performance of BackFlip to another recently released backbone flexibility predictor FlexPert-3D (Kouba et al., 2024). As discussed in Section 1.1 and Appendix A.1, FlexPert relies on the protein Language Model (pLM) embeddings and has a more complicated architecture with a trainable CNN head. We retrained BackFlip on the global RMSF metric and the dataset split used in FlexPert, both without and with one-hot encoded sequence embeddings, to also assess their effect. Without any sequence embedding, BackFlip outperforms FlexPert on both global and per-target Pearson correlation (as reported in the FlexPert paper), while performing slightly worse in terms of MAE (Tab. A.1). On the the *de novo* protein set, BackFlip significantly outperforms FlexPert on all metrics (Tab. A.2). We hypothesize that the reason for this might be that pLM embeddings are not informative for these proteins, as there is no evolutionary information available.

*Table A.1.* Comparison of performance of BackFlip to FlexPert on the ATLAS test set. BackFlip was retrained on the global RMSF metrics and the ATLAS dataset split used in FlexPert.

| Model | Global $r$ ↑ | MAE [Å] ↓ | Per-target $r$ ↑ | Per-target MAE [Å] ↓ |
|---|---|---|---|---|
| BackFlip* | 0.78 | 0.61 | **0.88** | 0.73 |
| BackFlip† | **0.81** | 0.56 | **0.88** | 0.72 |
| FlexPert‡ | 0.74 | **0.44** | 0.83 | **0.47** |

*No sequence embedding †One-hot sequence embedding embeddings ‡pLM embeddings

*Table A.2.* Comparison of performance of BackFlip to FlexPert on the set of 100 MD simulations of de novo proteins from Tab. 1. For inference with BackFlip, we use the same model checkpoint as in the Tab. A.1.

| Model | Global $r$ ↑ | MAE [Å] ↓ | Per-target $r$ ↑ | Per-target MAE [Å] ↓ |
|---|---|---|---|---|
| BackFlip[*] | **0.63** | **0.49** | **0.85** | **0.48** |
| FlexPert[‡] | 0.51 | 0.62 | 0.63 | 0.60 |

[*] No sequence embedding [‡] pLM embeddings

## A.6. Generation of the de novo protein dataset

We generate 20 protein backbones for each specified length $N \in [60, 65, \ldots, 512]$ using both FrameFlow (Yim et al., 2024) and RFdiffusion (Watson et al., 2023). Each generated backbone undergoes a self-consistency pipeline as introduced in prior work (Trippe et al., 2023; Watson et al., 2023; Yim et al., 2023b). This process involves designing eight sequences per backbone with ProteinMPNN (Dauparas et al., 2022) and refolding these sequences using ESMfold (Lin et al., 2023). We next analyze the length distribution of the ATLAS dataset (Vander Meersche et al., 2024), and select 50 backbones from each method (FrameFlow and RFdiffusion) that exhibit a self-consistency RMSD (scRMSD) of $\leq 2.0$ Å, and closely match the size distribution observed in the ATLAS dataset.

## A.7. Designability, novelty and diversity of protein backbones

In Tab. 1 we select 50 *designable* backbones sampled each with FrameFlow and RFdiffusion and in Tab. 2 we report *novelty* for backbones sampled for a certain flexibility profile. In order to find designable backbones, we follow a pipeline of self-consistency (Trippe et al., 2023; Watson et al., 2023) well-established in the protein design field. For each sampled backbone, we design 8 candidate sequences with ProteinMPNN (Dauparas et al., 2022), and predict their 3D structures with ESMfold (Lin et al., 2023), and define scRMSD as the smallest RMSD between our generated backbone and the 8 refolded, aligned candidates. Similar to (Watson et al., 2023; Yim et al., 2023b), we call backbone *designable* if the generated backbone has scRMSD $< 2$ Å.

We compute *novelty* analogously to (Wagner et al., 2024) by comparing the structures of protein backbones to the ones found in the PDB database using FoldSeek (Van Kempen et al., 2024) in TMalign mode (–alignment-type 1), and report *novelty* as the average of the highest TM score calculated for a given backbone, averaged over all proteins considered. *Diversity* from Tab. 3 is computed as a pairwise TM-similarity score for a set of protein backbones also as defined in (Wagner et al., 2024).

## A.8. Settings of Molecular Dynamics (MD) simulations

We conduct MD simulation using GROMACS v2023 (Abraham et al., 2015) with the all-atom force field CHARMM27. Since FliPS generates only protein backbones, we provide as input to MD simulation the structure of the best-scoring sequence refolded with ESMfold in terms of scRMSD. Proteins are placed in the periodic dodecahedron box with at least 1 nm distance from the box edge. The system is solvated with the TIP3P water model (Jorgensen et al., 1983) and NaCl concentration is adjusted to 150 mM. The system is energy-minimized for 5000 steps. NVT equilibration is performed for 1 ns with a 2 fs timestep using the leap-frog integrator with the temperature of 300K maintained with the Berendsen thermostat. Next, the system is NPT equilibrated for another 1 ns with the pressure set at 1 bar and maintained with the Parrinello-Rahman barostat. The production run is performed starting from the last frame of the NPT equilibration as three replicas of 100 ns resulting in the total simulation time of 300 ns. Covalent bonds involving hydrogen atoms were constrained using the LINCS (Hess, 2008) algorithm in all the simulations. Long range electrostatic interactions were accounted for using the Particle-Mesh Ewald (PME) method.

## A.9. Details on FliPS training

As mentioned in Section 4.2, we train FliPS on a subset of the PDB dataset. We observed that first pre-training on a smaller SCOPe dataset (Chandonia et al., 2022) annotated with BackFlip-predicted flexibilities helps stabilize training on the bigger PDB dataset and results in better performance. SCOPe can be considered as a benchmark dataset in the field of protein design, as it has been manually curated and used for training several generative models for protein backbones (Yim et al., 2023a; Lin & AlQuraishi, 2023). We filter SCOPe dataset clustered by 40 % sequence identitiy for lengths between 60 and

128 residues, which results in a set of 3673 structures. The hyperparameters are chosen as in GAFL (Wagner et al., 2024) and $\lambda_{\text{flex}}$ is set to 100. We train FliPS for a total of four GPU days on two NVIDIA A100 GPUs. Then, starting from this model checkpoint, we continue training on the filtered PDB dataset, as introduced in Section 4.2 for another 17 GPU days on 8 NVIDIA A100 GPUs, totalling 21 GPU days.

During training, 10% of the time we fully mask one-hot encoded per-residue flexibility and perform training of FliPS as an unconditional model with the original loss function introduced in (Yim et al., 2023a; Wagner et al., 2024). For remaining 90%, we randomly unmask flexibility as a window of size $\sim \mathcal{U}([0.2, 0.4])$ and with the window center positions $\sim \mathcal{U}([1, N])$ where $N$ stands for the total protein length. We do not use padding during the masking procedure.

### A.10. Sampling different protein lengths for a flexibility profile of interest with FliPS

In Section 4.2, we demonstrate that FliPS samples protein backbones of different lengths that display a flexibility profile given as an input. For this, we linearly scale the profile such that it has the same length as the generated sample.

### A.11. Definition of a local measure for flexibility

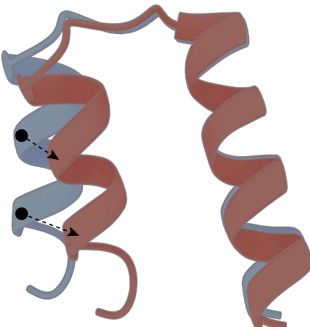

*Figure A.3.* A globally-aligned helix-turn-helix motif. Since the smaller helix has lower weight in the global alignment, it is assigned large (global) RMSF values although it is locally stiff. In dotted lines are the positions of the residues in the reference conformation.

The Root Mean Square Fluctuation (RMSF) commonly obtained either from conformational ensembles $\mathcal{C}$ of NMR-determined or MD-simulated protein structures is computed as

$$\text{RMSF} = \sqrt{\frac{1}{|\mathcal{C}|} \sum_{x \in \mathcal{C}} \left| \mathbf{T}^{\text{opt}} \circ x_i - x_i^{\text{ref}} \right|^2}, \tag{11}$$

with the reference conformation $x^{\text{ref}}$ and where $i$ denotes the residue index. The transformation $\mathbf{T}^{\text{opt}}$ is defined as

$$\mathbf{T}^{\text{opt}} = \underset{\mathbf{T} \in SE(3)}{\text{argmin}} \left\{ \sum_{i=1}^{N} \left( \mathbf{T} \circ x_i(t) - x_i^{\text{ref}} \right)^2 \right\} \tag{12}$$

where $N$ is the number of residues.

In simpler formulation, the RMSF shows the extent of positional deviation for a given residue from its reference state, all expressed in relation to the whole protein. Since the superposition in the definition of $\mathbf{T}^{\text{opt}}$ (Eq. 12) is a global quantity that depends on the whole protein, it is apparent that RMSF is *non-local* in the sense that the value of a given residue depends on the position of all other residues, even if their spatial distance is large. This suggests that the global RMSF cannot be the only measure for per-residue flexibility, as it often simply captures the movement of an entire, locally stiff subdomain with respect to the rest of the protein. This not only contradicts the notion of per-residue flexibility (as opposed to flexibility of subdomains or collective degrees of freedom), the non-locality can also lead to discontinuities. An example for such a case is illustrated in Fig. A.3A, where, due to the higher weight in the alignment, the longer helix in the helix-turn-helix

motif is rendered as stiff by the global RMSF and the other is rendered as flexible. If the shorter helix would be slightly longer, the global alignment would render it stiff and the other helix as flexible instead. For one and the same movement, the RMSF profile thus shows a large change upon a slightly change in the protein's topology. We not that this is no discontinuity in the strict mathematical sense since continuity cannot be defined for functions that depend on the length of individual subdomains because the length is discrete.

Addressing this issue, we propose a new, *local* definition of flexibility that avoids global superposition of the whole protein to the reference state in Section 2.2.

To illustrate the difference between local and global RMSF, we aligned 25 conformations of a 324 residues long protein (PDB: 1RM6) available from the ATLAS dataset according to the definition of local flexibility with sequence-window size $2K + 1 = 13$ and compare with standard global alignment (Eq. 12). Fig. A.4 demonstrates that in the case of local alignment, the secondary structure elements, such as $\alpha$-helices remain stable and well resolved, whereas in the case of global alignment, the helices experience significant positional changes and fluctuate more. As mentioned above, this will be manifested in the computation of RMSF, which will be significantly higher for the helical regions experiencing fluctuations in the global alignment.

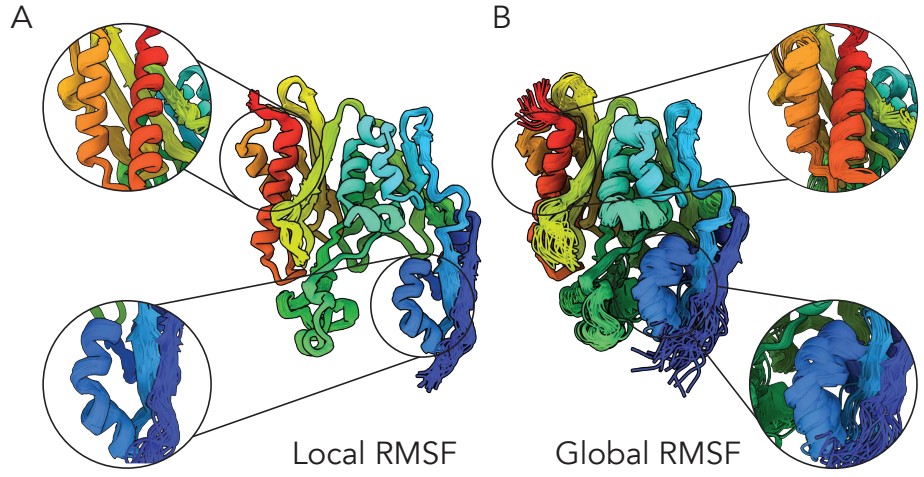

*Figure A.4.* Comparison of 25 conformations of a 324 residues long protein (PDB: 1RM6) aligned either locally, according to the definition of local flexibility in Eq. 4 (A) or globally (B). Bold circles highlight secondary structural elements where the difference between local and global alignment is most pronounced.

### A.12. Conditional FliPS sampling for proteins of bigger lengths

*Table A.3.* Flexibility conditioned backbone generation for 4 hand-drawn target profiles illustrated in the lower panel of Fig. 4. For each of the profiles, we generate backbones spanning lengths $N \in \{200, 215, \dots, 300\}$ using FliPS by conditioning on the flexibility profiles. See caption to Tab. 2 for other details.

| | MD of top samples | | | BackFlip on all samples | | |
|---|---|---|---|---|---|---|
| | $r$ ($\uparrow$) | MAE [Å] ($\downarrow$) | Novelty ($\downarrow$) | $r$ ($\uparrow$) | MAE [Å] ($\downarrow$) | Novelty ($\downarrow$) |
| **4 hand-drawn profiles** | | | | | | |
| FliPS | **0.79** (0.01) | **0.21** (0.00) | 0.61 (-) | **0.76** (0.00) | **0.27** (0.00) | 0.57 (0.01) |
| RFdiffusion-BFS | 0.41 (0.02) | 0.31 (0.00) | 0.57 (-) | -0.00 (0.00) | 0.32 (0.00) | 0.58 (0.01) |
| FoldFlow2-BFS | 0.35 (0.02) | 0.34 (0.00) | **0.48** (-) | -0.00 (0.00) | 0.33 (0.00) | **0.48** (0.00) |

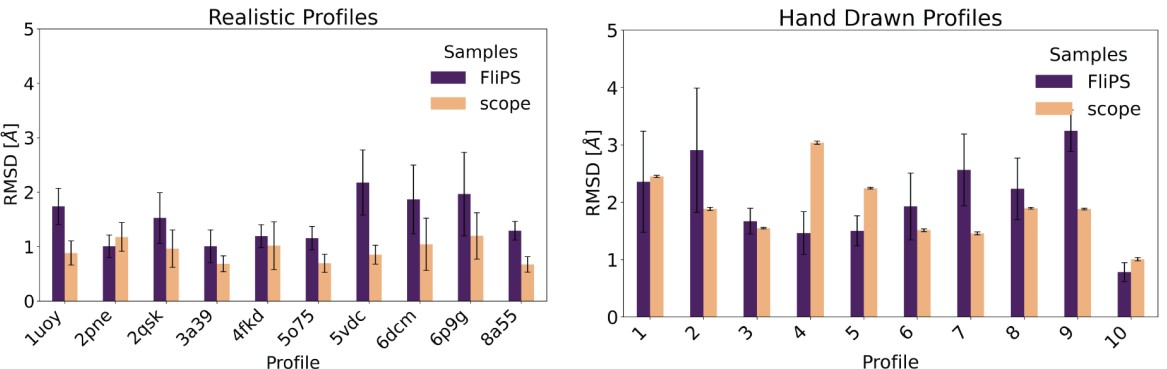

*Figure A.5.* Target profiles for benchmarking FliPS as in Tab. 2 and Tab. 3. (A) 10 realistic profiles (BackFlip-predictions of natural proteins outside the training set), (B) 10 hand-drawn profiles. Local RMSF computed from MD simulation of the generated samples is depicted in purple, and the target profile in green.

*Figure A.6.* scRMSD values of FliPS samples or naturally existing SCOPe proteins for a given set of flexibility profiles. Means and errors are computed as in Tab. 3 by bootstrapping the samples and selecting top 10-best ranked backbones 100 times.

## A.13. BackFlip guidance strategy and its ablation

We train FliPS model akin classifier-free guidance approach (Ho & Salimans, 2022), which requires the presence of interesting conditioning information at training time. We achieve this by flexibility-annotating the PDB dataset with BackFlip (Section 4.1) and passing flexibility profiles of the respective proteins to the model as a feature. The model then learns to approximate a conditional flow vector field (Eq. 6). A more general, training-free sampling strategy is classifier guidance (Dhariwal & Nichol, 2021), which combines the unconditional model score with a gradient term computed from a pre-trained classifier at test time. This allows to approximate the same conditional flow vector field as described above. Multiple studies have shown that classifier guidance-derived gradients might conflict with the unconditional generative direction (Dinh et al., 2023), which leads to non-convergence (Lou & Ermon, 2023) and quality-condition trade-off. We sought to investigate how classifier-guidance would compare to the FliPS model that is trained to directly approximate conditional flow vector field.

As described in Section 4.1, we implement BackFlip-guidance (BG) for flexibility-conditioned backbone generation by updating the prediction of the flow vector field made by unconditional model with a gradient computed by BackFlip w.r.t to intermediate structure $T_t$ at inference time $t$ (Eq. 9). We select the flexibility profile in the lower right corner in Fig. 4 as target and sample 100 backbones for each length $N \in \{200, 215, \ldots, 300\}$ using FliPS in unconditional model setting (see A.9) but with BG. For comparison, we also sample backbones unconditionally and using FliPS conditional model without BG. We assess the impact of various BackFlip-guidance (BG) parameters—including absolute BG scales (5 and 10), a linear scheduling scheme, and length-dependent scaling on performance across flexibility metrics (Section 4.2), secondary structure composition, and scRMSD of the generated backbones. In addition, we implemented BG within another unconditional model FrameFlow (Yim et al., 2024) and evaluated its performance with the same settings as described above.

Tab. A.4 reports results of the experiment. Indeed, application of BG significantly outperforms the unconditional sampling on flexibility metrics, but does underperform compared to FliPS sampling with conditional flow (FliPS*) regardless of the hyperparameters tested. We observe that the BG approach with a high absolute scale can significantly compromise backbone physical validity in favor of flexibility metrics, as indicated by high scRMSD values and illustrated in Fig. A.7. Although favorable flexibility metrics are achieved, there is a clear trend of over-representation of $\alpha$-helices when BG is applied. The optimal performance regards to both target flexibility, physical validity and secondary structure composition is achieved when the the effective BG scale is scaled with protein length. For FrameFlow-BG, we report inference results using the best-performing length-scaling schedule, and find similar performance when applying the same settings within the GAFL architecture. We also find that BG is approximately 20% slower than FliPS sampling with conditional flow.

*Table A.4.* Ablation of the BackFlip-Guidance (BG) strategy implemented either within FliPS or FrameFlow code. As the target we defined the flexibility profile in the lower right corner in Fig. 4 and sampled 100 backbones for each length $N \in \{200, 215, \ldots, 300\}$. A BG scale in linear schedule is computed as $\text{scale}(t) = \text{scale}_{\max} \cdot (t/t_{\max})$ with $\text{scale}_{\max}$ set as 5. We report metrics averaged over 10 top-ranked samples per profile. Standard deviations are computed by bootstrapping all generated samples 10 times before ranking them as described above. Unconditional flow means that the flexibility profile is not passed as a condition during inference. For more details see caption to Tab. 3.

|  | Med. scRMSD | $r$ ($\uparrow$) | MAE ($\downarrow$) | Helix pct. | Strand pct. | Coil pct. |
|---|---|---|---|---|---|---|
| Static schedule* | 2.78 (0.13) | 0.73 (0.01) | 0.30 (0.01) | 0.74 (0.13) | 0.05 (0.07) | 0.22 (0.02) |
| Static schedule† | 7.58 (0.53) | 0.78 (0.00) | 0.26 (0.01) | 0.65 (0.11) | 0.02 (0.03) | 0.33 (0.03) |
| Linear schedule | 1.60 (0.07) | 0.71 (0.01) | 0.31 (0.00) | 0.70 (0.15) | 0.07 (0.07) | 0.22 (0.02) |
| Length scaling | 1.80 (0.11) | 0.76 (0.00) | 0.25 (0.01) | 0.52 (0.20) | 0.07 (0.08) | 0.40 (0.04) |
| FrameFlow-length scaling | 3.23 (0.14) | 0.78 (0.01) | 0.25 (0.01) | 0.46 (0.09) | 0.08 (0.06) | 0.46 (0.02) |
| FliPS‡ | 1.15 (0.04) | 0.47 (0.05) | 0.33 (0.01) | 0.51 (0.24) | 0.21 (0.17) | 0.28 (0.03) |
| FliPS* | 2.78 (0.12) | **0.89** (0.00) | **0.24** (0.00) | 0.49 (0.08) | 0.19 (0.05) | 0.32 (0.00) |

*$\text{BG}_{\text{scale}} = 5$    †$\text{BG}_{\text{scale}} = 10$    ‡Unconditional flow    *Conditional flow

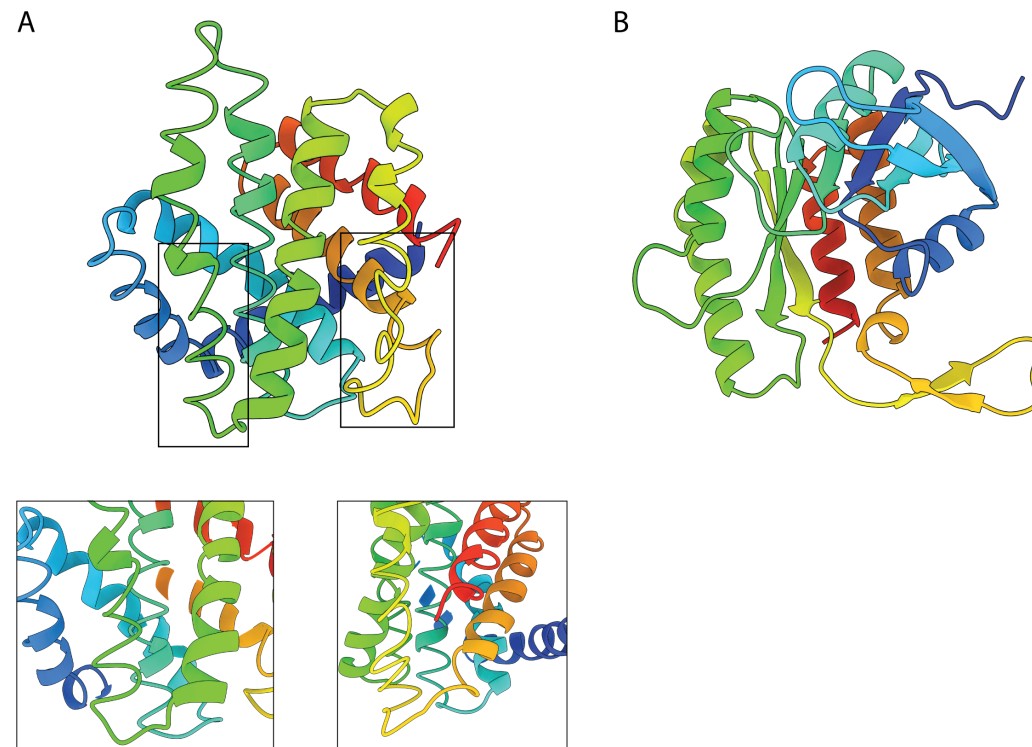

*Figure A.7.* (A) Example of a backbone generated with BackFlip guidance with a static schedule and BackFlip guidance scale set to 10 from Tab. A.4 with distorted, unphysical structure (scRMSD 15.1 Å). Especially prominent are improper torsional angles of the helix in green and a discontinuous helix in yellow. (B) Undesignable sample generated with FliPS using conditional flow with no BackFlip guidance (scRSMD 4.1 Å). The backbone does not display obvious physical violations or clashes, in contrast to (A).

