# OpenReview forum: "Flexibility-conditioned protein structure design with flow matching"
_ICML.cc/2025/Conference — ICML 2025 poster_

### Official Review · Reviewer_jPCN · 2025-03-10

**Overall Recommendation:** 3

**Summary:**

This paper introduces a novel framework for flexibility-conditioned protein structure design. The authors present BackFlip, an SE(3)-equivariant neural network that predicts per-residue flexibility from protein backbone structures. Using BackFlip, the authors propose GAFL-Flex, a flow matching-based generative model that conditions protein backbone generation on per-residue flexibility.

The authors evaluate both BackFlip and GAFL-Flex using molecular dynamics (MD), demonstrating that BackFlip accurately predicts structural flexibility, while GAFL-Flex successfully generates protein structures that exhibit dynamic behavior in accordance with target flexibility profiles.

**Claims And Evidence:**

1.BackFlip can predict per-residue flexibility from backbone structures without sequence information.

While BackFlip demonstrates strong performance, its evaluation is limited in scope. The authors primarily compare it against pLDDT, a confidence score from AlphaFold, and B-factors, which measure atomic displacement in X-ray crystallography and reflect thermal motion rather than intrinsic flexibility. Although pLDDT and B-factors correlate with flexibility, they are not standard flexibility benchmarks, raising concerns about the completeness of the validation. A more comprehensive evaluation could include alternative methods, such as using ProteinMPNN to generate multiple sequences for a given backbone, refolding them, and analyzing structural variability to see if BackFlip correctly predicts flexibility across different sequence designs. Additionally, the authors could apply BackFlip to NMR datasets, where proteins naturally exist in multiple conformations, providing a real-world test of its ability to capture experimentally observed flexibility.


2. GAFL-Flex can generate protein backbones that match desired flexibility profiles.

 While the results indicate that GAFL-Flex can generate structures with desired flexibility, the paper does not compare its performance against established baseline models in protein structure generation. RFdiffusion [1], FrameDiff, and other generative models have been shown to produce diverse and designable protein backbones, yet the paper does not examine whether GAFL-Flex actually generates more flexible structures than these existing approaches. A direct comparison with these models would provide stronger evidence that GAFL-Flex improves flexibility-aware design rather than simply being another generative method. The authors could conduct MD simulations on protein backbones generated by different models to assess whether GAFL-Flex produces statistically more flexible structures. Additionally, analyzing flexibility variations across multiple backbones from various generative models would further support the claim that GAFL-Flex introduces a meaningful improvement in flexibility control.


[1]Watson, Joseph L., et al. "De novo design of protein structure and function with RFdiffusion." Nature 620.7976 (2023): 1089-1100.
[2]Yim, Jason, et al. "SE (3) diffusion model with application to protein backbone generation." arXiv preprint arXiv:2302.02277 (2023).

**Essential References Not Discussed:**

NA

**Experimental Designs Or Analyses:**

As mentioned above, while the experiments are well-structured, they lack comprehensive baseline comparisons and real-world flexibility benchmarks. The evaluation of BackFlip is limited to comparisons with pLDDT and B-factors, which are correlated with flexibility but not standard benchmarks. A stronger validation would include NMR-derived ensembles to assess flexibility across experimentally observed conformations. Similarly, GAFL-Flex is not compared against existing generative models like RFdiffusion or FrameFlow, making it unclear whether flexibility conditioning provides a meaningful advantage. Expanding the evaluation with NMR datasets and alternative generative models would significantly strengthen the findings.

**Methods And Evaluation Criteria:**

The proposed methods and evaluation criteria do not fully align with the standard benchmarks used in protein flexibility prediction and generative protein structure modeling. While the authors demonstrate that BackFlip accurately predicts flexibility and that GAFL-Flex generates structures conditioned on flexibility, the evaluation remains incomplete due to the choice of baseline comparisons and the absence of key benchmark models.

For BackFlip, the authors validate its flexibility predictions using MD-derived local RMSF and compare it against AlphaFold’s pLDDT and B-factors from experimental data. However, while these metrics correlate with protein flexibility, they are not standard benchmarks for flexibility prediction. Although it is challenging to establish a proper baseline for a novel task—predicting flexibility solely from backbone structure—the authors should still include alternative baseline methods to provide a fair comparison. Additionally, a better validation would involve using NMR-derived ensembles, which capture experimentally observed conformational heterogeneity. NMR data provide multiple conformations of the same protein in solution, offering a real-world test of whether BackFlip can accurately predict backbone flexibility across different dynamic states.

Similarly, GAFL-Flex lacks a direct baseline comparison for protein structure generation. While the method is novel, structure generation is a well-established task, and comparing GAFL-Flex against existing models such as RFdiffusion and FrameFlow would provide stronger evidence of its effectiveness. The authors could also perform partial structure generation on NMR datasets, where they condition GAFL-Flex on rigid regions and generate flexible segments. This would allow them to test whether flexibility-conditioned generation outperforms naïve generative models in producing realistic flexible regions observed in experimental structures.

**Other Comments Or Suggestions:**

NA

**Other Strengths And Weaknesses:**

Strengths: The paper tackles an important and underexplored challenge in protein design—integrating flexibility into protein structure generation. This is crucial for designing proteins that target dynamic systems, such as antibodies and enzymes, where flexibility is essential for function.

Weaknesses: The study lacks baseline comparisons, making it unclear how GAFL-Flex compares to existing structure generation models. BackFlip's flexibility predictions are only evaluated against pLDDT and B-factors, which are not standard flexibility benchmarks.

**Questions For Authors:**

NA

**Relation To Broader Scientific Literature:**

This paper presents a well-motivated approach to rapidly evaluating protein dynamics and generating flexible protein structures, which addresses an important gap in the field of protein design. Current state-of-the-art protein design methods, such as those used for binder design, have achieved significant success when targeting static structures. However, for flexible targets, the success rate is notably lower, as existing generative models do not explicitly account for conformational dynamics. By introducing flexibility-aware protein structure generation, this work has the potential to impact areas such as antibody design, enzyme engineering, and intrinsically disordered protein modeling.

**Theoretical Claims:**

The paper does not focus on formal theoretical claims or proofs but builds on established principles from flow matching, geometric deep learning, and SE(3)-equivariant networks.

---

> ### Author Rebuttal · Authors · 2025-03-31
>
> We thank the reviewer for their constructive and helpful review. We are happy the reviewer finds the problem of flexibility-conditioned design important. Due to character constraints, we try to focus on the most important concerns below.
>
> - We note that we retrained GAFL-Flex on the larger PDB dataset and observe enhanced performance at the original benchmark (answer to 3jah, section 'Evaluation of the model trained on the PDB').
>
> - We also note that we introduce a novel BackFlip-guidance approach for conditional generation that we evaluate on longer proteins (answer to 3jah, section 'General response' and 'Experiments on longer proteins').
>
> i. **BackFlip can predict per-residue flexibility without sequence information**
>
> **General comment on the scope of BackFlip as MD-flexibility-emulator**
>
> We first want to emphasize that BackFlip serves as a speed-up for MD simulations, which are prohibitively expensive for dataset annotations, and can thus be seen as an MD-flexibility-emulator. Thus, by learning to predict flexibility derived from MD simulations, BackFlip inherits biases of and correlations between ground-truth MD-derived flexibility and any experiment-derived flexibility.
>
> **Scope of the comparison of BackFlip is limited to B-factors and pLDDT**
>
> Crystallographic B-factors and pLDDT are widely regarded as metrics of flexibility and we regard showing that they do not correlate well to MD as important.
>
> We include another recent flexibility prediction model, FlexPert [1], which combines embeddings from a large protein language model (pLM) with structural features, as a baseline. Similar to BackFlip, FlexPert is trained on the ATLAS dataset. We retrained BackFlip on the global RMSF metric and the dataset split used in FlexPert. **For the results, we refer to the Tables R3, R4 in the response to the reviewer VDWk.** BackFlip without sequence embedding is better or on-par with the sequence-informed, larger model FlexPert on the ATLAS test set and outperforms it on the set of MD simulations of 100 de novo proteins. We conclude that BackFlip is the current SOTA model for predicting MD-derived flexibility.
>
> **Comparison of BackFlip to NMR-derived flexibility**
>
> We thank the reviewer for their suggestion. As discussed above, BackFlip solves the task of predicting MD-derived flexibility and predicting NMR-derived flexibility is out of scope. We expect BackFlip to inherit the upsides and downsides of MD, i.e. also the correlation of MD- to NMR-derived flexibility. However, we did apply BackFlip to 500 randomly selected NMR ensembles from the BMRB database [2] and compared it to other baselines, where it achieves SOTA performance as well and outperforms FlexPert, in particular. Table R5 summarizes the results.
>
> We find it is important to note, while NMR structures are commonly deposited as conformational ensembles, these often represent averages over heterogeneous states over broad time spans and cannot be seen as true statistical ensembles [3]. Even the (very) recent micro-milisecond dynamics predictor Dyna-1 [4], trained directly on spin relaxation times observed in NMR, correlates very poorly with the flexibility of PDB-deposited NMR ensembles (Table R5, Dyna-1).
>
> **Table R5: Performance of BackFlip on RMSF prediction of 500 randomly selected NMR ensembles.**
>
> |**Model**|**Global *r* (↑)**|**Global MAE (↓)**|
> |-|-|-|
> |Negative pLDDT|0.45| -|
> |FlexPert|0.43|2.1|
> |Dyna-1| 0.16| -|
> |**BackFlip**|**0.65**|**2.0**|
>
> ii. **Comparison of GAFL-Flex with SOTA structure generative models**
>
> We followed the reviewer's suggestion to compare GAFL-Flex with other baselines. We chose as baselines RFdiffusion and FoldFlow2 [5], as these two models demonstrate SOTA performance for unconditional generation.
>
> We retrained GAFL-Flex on a BackFlip-annotated PDB dataset. We compare this new model (GAFL-Flex\*) with the baselines and find that conditional sampling results in proteins that, on average, more closely follow the desired flexibility profile and that are significantly more flexible compared to proteins sampled unconditionally using the baselines. The results and more details on the experiments can be found in (answer to 3jah, section 'Experiments on longer proteins').
>
> ---
> **References**
>
> [1] Kouba, Petr, et al. "Learning to engineer protein flexibility." arXiv:2412.18275 (2024).
>
> [2] Hoch, Jeffrey C., et al. "Biological magnetic resonance data bank." Nucleic acids research 51.D1 (2023): D368-D376.
>
> [3] Bonomi, Massimiliano, et al. "Principles of protein structural ensemble determination." Current opinion in structural biology 42 (2017).
>
> [4] Wayment-Steele, Hannah K., et al. "Learning millisecond protein dynamics from what is missing in NMR spectra." bioRxiv (2025).
>
> [5] Huguet, Guillaume, et al. "Sequence-augmented se (3)-flow matching for conditional protein backbone generation." arXiv:2405.20313 (2024).

---

### Official Review · Reviewer_7Wxe · 2025-03-13

**Overall Recommendation:** 2

**Summary:**

This paper takes a step towards overcoming this limitation by proposing a framework to condition structure generation on flexibility, which is crucial for key functionalities such as catalysis or molecular recognition. The authors first introduce BackFlip, an equivariant neural network for predicting per-residue flexibility from an input backbone structure. Relying on BackFlip, we propose GAFL-Flex, an SE(3)-equivariant conditional flow matching model that solves the inverse problem, that is, generating backbones that display a target flexibility profile.

**Claims And Evidence:**

This paper introduces a generative model for protein structure design conditioned on per-residue flexibilities using flow matching.

The experiments demonstrate that flexibility-conditioning leads to the generation of diverse and novel backbones that indeed display the respective target flexibility profile in Molecular Dynamics simulations.

**Essential References Not Discussed:**

The most related work, Dynamics-Informed Protein Design with Structure Conditioning (ICLR2024) needs a discussion, and comparison is appreciated.

**Experimental Designs Or Analyses:**

The experiments demonstrate that flexibility-conditioning leads to the generation of diverse and novel backbones that indeed display the respective target flexibility profile in Molecular Dynamics simulations.

**Methods And Evaluation Criteria:**

This paper introduces a generative model for protein structure design conditioned on per-residue flexibilities using flow matching. The proposed flexibility-conditioning framework relies on the structure-based flexibility prediction model BackFlip, enabling large scale flexibility annotation of proteins, a novel flexibility auxiliary loss and a flexibility screening procedure to find protein backbones that best display a flexibility profile of interest.

This paper also proposes a generalization of RMSF as Local RMSF, in which the fluctuations of a residue are measured with respect to its local surrounding instead of the whole protein. This paper evaluate the flexibility with MD.

**Other Comments Or Suggestions:**

More advanced conditioning techniques for steering (protein) dynamics are appreciated.

**Other Strengths And Weaknesses:**

Conditioning on protein dynamics is a promising direction, further elaborations (e.g., case study about the application) on this are appreciated.

**Questions For Authors:**

Could this per-residue dynamics reflex some specific patterns of protein motifs as in Dynamics-Informed Protein Design with Structure Conditioning (ICLR2024) ？

**Relation To Broader Scientific Literature:**

This is a direct application of Riemannian Flow Matching in Protein design.

**Theoretical Claims:**

There is no significant theoretical claims.

---

> ### Author Rebuttal · Authors · 2025-03-31
>
> We thank the reviewer for their time invested in reading the paper and for their constructive feedback. We are happy that the reviewer agrees with our claims and evidence and appreciates the methods and evaluation criteria and our experimental design. Below we will discuss the comments line by line.
>
> - We note that we retrained GAFL-Flex on the larger PDB dataset and observe enhanced performance at the original benchmark (answer to 3jah, section 'Evaluation of the model trained on the PDB').
>
> - We also note that we introduce a novel BackFlip-guidance approach for conditional generation that we evaluate on longer proteins (answer to 3jah, section 'General response' and 'Experiments on longer proteins').
>
> i. **Dynamics-Informed Protein Design with Structure Conditioning**
>
> We thank the reviewer for pointing us toward this work [1]. We have added a discussion in the related work section:
>
> (U. Komorowska et al., 2024) propose conditioning a pre-trained diffusion model on normal modes—i.e., Hessians of the potential energy predicted by a force field that approximate local movements around an equilibrium state. Conditioning is achieved via inference-time guidance, using gradients computed from an analytical normal mode loss.'
>
> While the approach may appear similar to the flexibility-conditioning proposed in our paper, both the task and the method are fundamentally different:
>
> 1. We propose training a *conditional* model: akin to classifier-free guidance [2], we pass the condition as input and learn a conditional flow. In contrast, [1] uses an *unconditional* diffusion model guided during inference by gradients from an analytical scoring function, following a classifier guidance scheme.
>
> 2. The conditioned quantity differs: we use flexibility derived from Molecular Dynamics (MD) simulations, while [1] relies on Normal Mode Analysis (NMA)—a simulation-free approximation that assumes a harmonic potential and is only valid near equilibrium.
>
> 3. The guidance approach in [1] requires an *analytical* condition to compute gradients. In contrast, our flexibility-conditioning approach can handle *non-analytical* conditions (e.g., MD-derived flexibility) because we train the model to approximate a conditional flow, rather than apply analytical gradient guidance.
>
> Since the models condition on different quantities (analytical NMA vs. MD-derived flexibility), they solve different tasks and are not directly comparable. While GAFL-Flex can accept any flexibility profile (e.g., derived via NMA), making it a general method for dynamics-informed design, the approach in [1] is limited by its reliance on the harmonic assumption in NMA, which may not always hold.
>
> **Q: Could this per-residue dynamics reflect specific patterns of protein motifs?**
>
> Since lowest non-trivial modes in [1] are computed for the entire protein structure, and motifs are subsequently sampled as sub-regions of these structures, it can indeed be expected that MD-derived flexibilities will correlate with the amplitudes of lowest non-trivial modes of oscillations from NMA.
>
> ii. **More advanced conditioning techniques for steering dynamics**
>
> We retrained GAFL-Flex on BackFlip-annotated PDB of monomeric structures (22977) and developed BackFlip guidance in analogy to classifier guidance. With BackFlip-guidance, we can achieve the same flexibility-similarity as with the conditional model on the four new flexibility profiles for longer proteins, however, it is around 20 percent slower and requires more memory. Training on the larger dataset improves the conditioning performance and yields more novel backbones. For more details we refer to the answer to 3jah, sections 'General response' and 'Experiments on longer proteins'.
>
> iii. **Further elaborations on flexibility-conditioned design**
>
> We also regard this as promising direction and plan to use the framework introduced in this paper in more application-related work in the future. For instance, structural flexibility is believed to be important for the functionality of protein assemblies [3], for enzymatic catalysis [4] and for binding events of flexible receptor and respective ligand-proteins [5].
>
> **Note: We also evaluated another recent flexibility prediction model and observe that BackFlip outperforms it (see answer VDWk, Table  R3, R4, Section i).**
>
> ---
> **References**
> [1] U. Komorowska et al. Dynamics-Informed Protein Design with Structure Conditioning. ICLR 2024
>
> [2] Ho & Salimans, 2022 – Classifier-Free Guidance
>
> [3] Khmelinskaia, Alena, et al. "Local structural flexibility drives oligomorphism in computationally designed protein assemblies." Nature Structural & Molecular Biology (2025): 1-11.
>
> [4] Matsuo, Takashi, et al. "Global structural flexibility of metalloproteins regulates reactivity..." Chemistry–A European Journal 24.11 (2018): 2767-2775.
>
> [5] Craveur, Pierrick, et al. "Protein flexibility in the light of structural alphabets." Frontiers in molecular biosciences 2 (2015): 20.

---

### Official Review · Reviewer_VDWk · 2025-03-13

**Overall Recommendation:** 3

**Summary:**

This paper introduces a novel framework for de novo protein design that explicitly incorporates residue-level flexibility—a dynamic property critical for biological function—into the generative process. Current methods prioritize static structural features (e.g., motifs, symmetry), limiting their ability to engineer proteins for dynamic processes like catalysis. The authors address this gap with two key innovations: BackFlip, an SE(3)-equivariant network predicting flexibility from backbone structures, and GAFL-Flex, a conditional flow model that inversely generates backbones conditioned on target flexibility profiles. Extensive experiments show that BackFlip can accurately predict the flexibility with a high Pearson correlation coefficient on unseen data and GAFL-Flex can generate plausible proteins conditioned on target flexibility profile. The paper is overall well-written.

**Claims And Evidence:**

This paper claims that "Back-Flip is a backbone flexibility predictor which is entirely independent of sequence information". However, flexibility is intrinsically coupled with sequence as side-chain type and conformation are keys to protein thermostability and flexibility. Therefore, my concern is that flexibility can not be accurately predicted without any sequence information as an input. I kindly suggest authors can elaborate on this point.

**Essential References Not Discussed:**

I did not find.

**Experimental Designs Or Analyses:**

The experimental design raises concerns regarding the role of side chains in evaluating protein dynamics. While protein flexibility depends on both backbone and side-chain interactions, the proposed approach relies solely on backbone MD simulations to validate the results in Table 2o, which introduces uncertainty. A direct comparison of flexibility between backbone-only and full-atom MD (e.g., using the Atlas dataset) is necessary to assess whether backbone-only simulations sufficiently capture dynamic behavior or if side-chain contributions are critical. Additionally, there is a potential inconsistency in dataset usage—the Back-Flip model is trained on the Atlas dataset, which includes side-chain information, yet the generative module is validated using backbone-only MD simulations. If Back-Flip is trained properly, it predicts all-atom flexibility, which should not be directly compared to de novo protein backbone flexibility. Addressing these discrepancies through additional controlled experiments would strengthen the validity of the approach.

**Methods And Evaluation Criteria:**

The proposed method of training a backbone flexibility predictor followed by conditioned generation model makes sense to me.

**Other Comments Or Suggestions:**

I have no further questions for authors. I will keep a positive rating if my concern can be well addressed.

**Other Strengths And Weaknesses:**

No other strengths and weaknesses.

**Questions For Authors:**

## Update after rebuttal
I thank authors for thier detailed responses and I would like to keep my positive rating to accept this paper.

**Relation To Broader Scientific Literature:**

Traditional protein modeling tools such as Rosetta MotifGraft (Alford et al., 2017) with BackRub sampling (Lauck et al., 2010), Modeller (Šali & Blundell, 1993), and LoopGrafter (Planas-Iglesias et al., 2022) enable flexibility engineering in specific regions through structure-guided iterative sampling using empirical energy functions, but their applicability is constrained by reliance on predefined input structures and high computational costs. While hybrid approaches combining classical tools and deep learning have successfully designed allosteric proteins (Pillai et al., 2024), pH-responsive complexes (Shen et al., 2024), and fold-switching systems (Guo et al., 2024), current deep learning methods primarily focus on structure prediction rather than generative target structure design and lack explicit flexibility conditioning. In concurrent work, Kouba et al. (2024) propose FlexPert-3D, a sequence-based pipeline using molecular dynamics-derived flexibility to fine-tune ProteinMPNN (Dauparas et al., 2022) via evolutionary priors from protein language models. However, their framework operates solely in sequence space and depends on evolutionary information, fundamentally contrasting with our structure-centric generative approach that directly encodes flexibility into structural design.

**Theoretical Claims:**

This paper does not claim theoretical contributions, so there are no theoretical claims. I have checked the formulas used in this paper. They are correct and understandable.

---

> ### Author Rebuttal · Authors · 2025-03-31
>
> We cordially thank the reviewer for their time invested in reading the paper and for their constructive review. We discuss questions and concerns below line by line.
>
> - We note that we retrained GAFL-Flex on the larger PDB dataset and observe enhanced performance at the original benchmark (answer to 3jah, section 'Evaluation of the model trained on the PDB').
>
> - We also note that we introduce a novel BackFlip-guidance approach for conditional generation that we evaluate on longer proteins (answer to 3jah, section 'General response' and 'Experiments on longer proteins').
>
> i. **Backbone flexibility prediction in absence of sequence information**
>
> We compared the performance of BackFlip to FlexPert [1], another backbone flexibility predictor that combines embeddings from a large protein language model (pLM) with structural features. Similar to BackFlip, FlexPert is trained on the ATLAS dataset. We retrained BackFlip on the global RMSF metric and dataset split used in FlexPert, both without and with one-hot encoded sequence embeddings, to assess their effect.
>
> Table R3 reports inference results on the ATLAS test set. Without any sequence embedding, BackFlip outperforms FlexPert on both global and per-target Pearson correlation (as reported in the FlexPert paper), while performing slightly worse in terms of MAE. Indeed, one-hot encoded sequence improves the performance of BackFlip, but it is clearly possible to estimate the flexibility already from the structure alone. Since we utilize BackFlip in the auxillary loss during training of GAFL-Flex and screening of de novo generated backbones, where the sequence is not defined, we find it advantageous that BackFlip demonstrates such a strong performance without requiring any sequence as input. We regard this as an important contribution since it contrasts the paradigm of relying solely on evolutionary or sequence information for predicting dynamical properties.
>
> We also compared BackFlip and FlexPert on a set of de novo proteins (from Table 1 in the main text of the submission), with results shown in Table R4. BackFlip significantly outperforms FlexPert on all metrics. We think the reason for this might be that pLM embeddings are not informative for these proteins, as there is no evolutionary information available.
>
> These results support our hypothesis that the geometry of a backbone and secondary structure composition of a well-folded, globular protein are sufficient to infer short-range nanosecond backbone flexibility without sequence information. However, we agree with the reviewer that sequence information is more important when it comes to long-range protein flexibility. We expect that BackFlip will generalize worse to highly dynamical systems, such as intrinsically disordered proteins, where dynamics is in the range of micro or milliseconds and is dictated to a large extent by the sequence [2].
>
> ii. **On MD simulations of generated de novo backbones**
>
> We apologize for any lack of the clarity on MD procedure. We will clarify this in the final version.
>
> We conduct all MD simulations following the protocol published in the ATLAS paper, that is, all-atom simulations for 300 ns conducted as 3 replicas with explicit TIP3P water as solvent. The input structure to the MD simulation is the ESMfold-refolded protein (sequence is designed by ProteinMPNN, see A.7 section in the paper) with the lowest scRMSD to the backbone generated by GAFL-Flex. We report all metrics (r and MAE) on Cα RMSF profiles, but these are extracted from the all-atom trajectories.
>
> Indeed, BackFlip is trained to predict Cα RMSF from the ATLAS dataset structures, which are all-atom. Accordingly, all evaluations report Cα RMSF metrics. BackFlip only sees [N, CA, C] backbone atoms during training, thus it only implicitly predicts effects of side chain atom interactions - like AlphaFold2.
>
> **Table R3: Performance of BackFlip retrained with or without sequence embedding on the global RMSF metric on ATLAS dataset split of FlexPert [1].**
>
> | Model|Global *r* (↑)|MAE [Å] (↓)|Per-target *r* (↑)|Per-target MAE (↓)|
> |-|-|-|-|-|
> |BackFlip *|0.78|0.61|0.88|0.73|
> |BackFlip †|0.81|0.56|0.88|0.72|
> |FlexPert ‡|0.74|0.44|0.83|0.47|
>
> * No sequence embedding † One-hot sequence embedding ‡ pLM sequence embedding
>
> **Table R4: Performance of BackFlip without sequence embedding retrained on the global RMSF metric on ATLAS dataset split of FlexPert [1] on MD simulations of 100 de novo proteins sampled with RFdiffusion or FrameFlow.**
>
> |Model| Global *r* (↑)|MAE [Å] (↓)|Per-target *r* (↑)|Per-target MAE [Å] (↓)|
> |-|-|-|-|-|
> |BackFlip *|0.63|0.49|0.85|0.48|
> |FlexPert ‡|0.51|0.62|0.63|0.60|
>
> * No sequence embedding † One-hot sequence embedding ‡ pLM sequence embedding
>
> ---
> **References**
>
> [1] Kouba, Petr, et al. "Learning to engineer protein flexibility." arXiv preprint arXiv:2412.18275 (2024).
>
> [2] Radivojac, Predrag, et al. "Protein flexibility and intrinsic disorder." Protein Science 13.1 (2004): 71-80.

---

> > ### Comment · Reviewer_VDWk · 2025-04-02
> >
> > I thank authors for the detailed responses and I would like to keep my positive rating to accept this paper.

---

> > > ### Author Response · Authors · 2025-04-03
> > >
> > > We are glad the reviewer has a positive view of the paper and recommends its acceptance. We are happy to answer any further open questions!

---

### Official Review · Reviewer_3jah · 2025-03-14

**Overall Recommendation:** 2

**Summary:**

This paper proposed a framework for conditional structure generation conditioning on desired flexibility, a key characteristic in catalytic interactions and molecular recognition. They develop BackFlip, a backbone flexibility predicter that can be used for large-scale flexibility annotation, and combine it with a Geometric Algebra Flow Matching model to achieve flexibility conditioned generation.
They show that GAFL-Flex can generate novel protein backbones with the desired flexibility, verified by Molecular Dynamics (MD) simulations.

**Claims And Evidence:**

The claim that BackFlip can accurately predict flexibility profile (as measured by MD RMSF) is supported by experimental results onthe  ATLAS test dataset. Although the true flexibility might be different from the MD simulation.
The flexibility-conditioned generation performance is supported by some evidence from small proteins, but its capability to generalize to bigger proteins (that are likely to contain more flexible regions) is undetermined.

**Essential References Not Discussed:**

The references are covered quite comprehensively.

**Experimental Designs Or Analyses:**

As mentioned in the Methods and Evaluation criteria, experimental designs are valid. It would be nice if a comparison with strong unconditional models, as well as classifier-guidance results, could be added (e.g., using DPS with BackFlip prediction as objective).

In addition, experiments on longer proteins (>128 aa) are needed to prove that the model can be scaled to more complex protein backbones (and likely to contain more coils).

**Methods And Evaluation Criteria:**

The authors proposed to use local RMSF instead of global RMSF to quantify flexibility and generate 10 conformations from MD to produce ground-truth training data. The conditional flow matching model is trained using similar techniques in classifier-free guidance that balances conditional and unconditional training and uses auxiliary loss that penalizes the deviating (predicted) flexibility of the generated structure.  The methodology overall makes sense, although the training data seems to focus only on short proteins with len 60-128 (3673 structures) and might be too small to reliably learn complex geometries of the protein backbones.

Since the BackFlip score is differentiable, maybe another baseline they should compare is directly using BackFlip to do test time guidance of pre-trained large-scale backbone generative models such as RFDiffusion.

**Other Comments Or Suggestions:**

No other comments.

**Other Strengths And Weaknesses:**

It seems comparison on pLDDT and RMSD is omitted in the evaluation as the authors thought that more flexible protein will lead to worth RMSD/pLDDT. However, since both metrics are still important in current de novo design, how should we measure the designability/foldability of flexibility-conditioned designs if these two metrics lose their meanings?

**Questions For Authors:**

Does the model scale to larger data and longer proteins?
Do we know if the model is not solely recognizing alpha-helix and beta sheet? Are there quantitative measures on how well the model differentiate flexibilities in non-loop regions?

**Relation To Broader Scientific Literature:**

The paper is one of the first method that this reviewer know on flexibility conditioned design. The task can be generally related to molecular dynamics generation of proteins, such as MD trajectory learning, conformation sampling, and structure prediction.

**Theoretical Claims:**

The paper is mostly method development and empirical evaluation and therefore does not have theoretical results to assess.

---

> ### Author Rebuttal · Authors · 2025-03-31
>
> We thank the reviewer for the time they invested in reading the paper and their helpful suggestions!
>
> i. **General response**
>
> We retrained GAFL-Flex on the BackFlip-annotated PDB dataset of 22977 monomeric protein structures filtered by the (i) length between 60 and 512 residues and (ii) absence of breaks in the structure, and conducted a series of new experiments based on the review. The resulting model generates protein backbones that better match the desired flexibility profiles, demonstrates improved novelty, and succeeds at designing larger proteins for challenging flexibility profiles. We developed a BackFlip-guidance (BG) approach similar to classifier guidance that performs well but is slower than the original conditional model. Due to time constraints, we applied it with our unconditional model but will extend it to RFdiffusion in the final submission. We discuss new experiments below.
>
> **Evaluation of the model trained on PDB**
>
> We re-ran the experiment from the main text of our submission reported in the Table 2 and found that GAFL-Flex trained on PDB performs better than GAFL-Flex trained on SCOPe in terms of correlation and yields more novel backbones (Table R1).
>
> **Table R1: Performance of GAFL-Flex trained on the PDB (GAFL-Flex\*) at the benchmark reported in Table 2 of the original submission.**
>
> ||r (↑)|MAE [Å] (↓)|Novelty (↓)|
> |-|-|-|-|
> |**10 existing profiles**|||||
> |GAFL-Flex\*|**0.52 (0.00)**|0.20 (0.00)|**0.64 (0.02)**|
> |GAFL-Flex|0.45 (0.00)|**0.17 (0.00)**|0.68 (0.02)|
> |GAFL-uncond.|0.20 (0.00)|0.20 (0.00)|0.73 (0.02)|
> |SCOPe proteins|0.19 (0.00)|0.25 (0.00)|1.0 (-)|
> |**10 arbitrary profiles**|||||
> |GAFL-Flex\*|**0.56 (0.00)**|0.44 (0.00)|**0.64 (0.02)**|
> |GAFL-Flex|0.47 (0.00)|**0.43 (0.00)**|0.68 (0.03)|
> |GAFL-uncond.|0.09 (0.00)|0.48 (0.00)|0.72 (0.02)|
> |SCOPe proteins|0.09 (0.01)|0.48 (0.00)|1.0 (-)|
>
> ii. **Experiments on longer proteins**
>
> We defined 4 new target flexibility profiles suitable for longer proteins (given by 3 to 5 rectangular peaks with widths of 10 to 20 residues and amplitudes of 1 to 2.5 Å). We sampled 100 protein backbones for each length in [200, 250, 300]. We also included the SOTA unconditional structure generative models RFdiffusion and FoldFlow2 [1].
>
> Table R2 reports the results of the experiment. Conditional generation yields protein backbones that closely follow the respective target profiles. Unconditional sampling, on the contrary, produces samples that do not reflect desired profiles. Similar to the experiment from our submission, we observe that conditioning improves novelty of sampled backbones. Average flexibility is elevated compared to unconditional generation.
>
> Remarkably, BackFlip-guidance (GAFL-BG\*) achieves the same performance as the conditional model (GAFL-Flex\*). GAFL-BG is about 20% slower.
>
> **Table R2: Performance of GAFL-Flex for longer proteins. We evaluated 4 new target flexibility profiles. Sampled lengths L ∈ [200, 250, 300], each 100 backbones. Metrics are evaluated using BackFlip on all samples.**
>
> | Method| r (↑)| MAE [Å] (↓)| Novelty (↓)| Avg. Flex [Å]|
> |-|-|-|-|-|
> | **4 arbitrary profiles**|||||
> | GAFL-Flex\*|**0.757**|**0.268**|0.569| 0.730|
> | GAFL-BG\*|**0.757**|**0.267**|0.566|0.730|
> | GAFL-uncond.\*|-0.03|0.33| 0.67|0.56|
> | FoldFlow2|0.00|0.33|**0.48**|0.46|
> | RFdiffusion|-0.01|0.32| 0.58|0.55|
>
> iii. **Designability of flexibility-conditioned generated backbones**
>
> We observe that the more novel and flexible even natural protein backbones are, the higher the self-consistency RMSD (scRMSD) computed in the refolding pipeline becomes (Figure 4 in the main text of the submission). Designability depends on the target flexibility profile (Figure A.8), which is not surprising as flexibility introduces fundamental uncertainty to scRMSD. We regard our finding as an important contribution to rethinking the designability definition for flexible protein design. We think approaches for alternative definitions could include making the refolding-RMSD threshold dependent on novelty and flexibility or to only consider stiff parts for calculating scRMSD.
>
> If one would rely on the well-established cut-offs for designability (e.g. scRMSD < 2.0, pLDDT > 70), one would inevitably introduce a bias towards selecting rigid proteins. It is important to note that these cutoff values were conceived having static protein representation in mind.
>
> iv. **Distinction between flexibilities in structured regions**
>
> Due to time constraints during the rebuttal phase, we were not able to make an experiment in this regard. However, we believe this is a great suggestion and we think can answer this question by computing the Pearson correlation and MAE to the ground-truth MD flexibility by masking loops during the computation of metrics. We will include this analysis in the final version of the paper.
>
> **Note: We also evaluated another recent flexibility prediction model and observe that BackFlip outperforms it (see answer VDWk, Table  R3, R4, Section i).**

---

### Decision · Program_Chairs · 2025-05-01

**Decision:**

Accept (poster)

**Comment:**

This paper presents a novel and well-motivated framework for protein structure generation conditioned on per-residue flexibility—an important but under-explored aspect of protein design. The authors introduce BackFlip, a backbone-only model for predicting flexibility from MD data, and GAFL-Flex, a conditional flow model that generates backbones to match target flexibility profiles. The approach is validated through comprehensive experiments using Molecular Dynamics simulations.

The paper tackles a timely challenge: designing proteins with dynamic, functional properties such as flexibility. The proposed models are thoughtfully designed, and the results demonstrate that GAFL-Flex can generate novel, flexible backbones aligned with specified profiles.

Reviewers raised concerns about generalization to longer proteins, comparisons to baselines like RFdiffusion, and whether flexibility can be predicted without sequence input. The authors responded with new experiments, retraining models on larger datasets, benchmarking against strong baselines, and evaluating on NMR data. These additions substantially strengthen the paper and clarify its contributions.

While not all reviewers updated their scores, the rebuttal clearly addressed the main concerns with compelling evidence. This work lays a strong foundation for future research in dynamics-aware protein design.

I recommend acceptance.